# To Sink or Not to Sink: Visual Information Pathways in Large Vision-Language Models

**Jiayun Luo**[1,2]*   **Wan-Cyuan Fan**[1,2]*   **Lyuyang Wang**[1]   **Xiangteng He**[1,2]

**Tanzila Rahman**[1,2]   **Purang Abolmaesumi**[1]   **Leonid Sigal**[1,2]

[1]**University of British Columbia**   [2]**Vector Institute for AI**
{letitial, wancyuan}@cs.ubc.ca

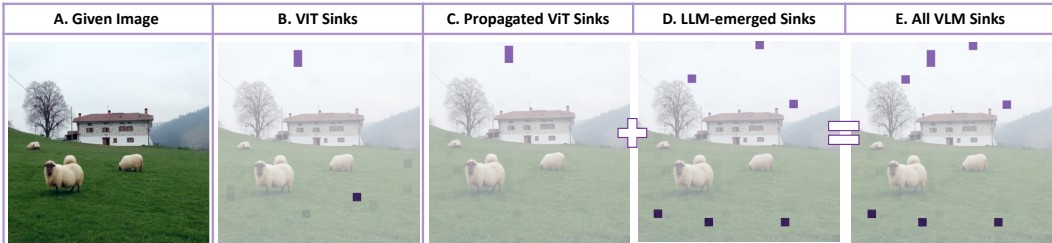

| A. Given Image | B. VIT Sinks | C. Propagated ViT Sinks | D. LLM-emerged Sinks | E. All VLM Sinks |
|---|---|---|---|---|

Figure 1: **Illustration of ViT and LLM attention sinks in LLaVA-v1.5-7B.** In LVLMs, given an image (A), we find that ViT sinks (B) are partially propagated into the LLM as (C), alongside LLM-emerged sinks (D), together outlining all sinks within the VLM (E).

## Abstract

Large Vision Language Models (LVLMs) have recently emerged as powerful architectures capable of understanding and reasoning over both visual and textual information. These models typically rely on two key components: a Vision Transformer (ViT) and a Large Language Model (LLM). ViT encodes visual content into a sequence of image tokens and serves as the perceptual front-end – the *eyes* of the model. In contrast, the LLM interprets these tokens to perform high-level reasoning, generates responses, and functions as the cognitive core – the *brain* of the model. However, it remains unclear which visual tokens contribute most significantly to understanding and reasoning, and how effectively these signals are propagated from ViT to the LLM. While most existing works have focused on identifying *attention sinks*, low-semantic tokens receiving disproportionately high attention, within the LLM, we shift the focus to the vision encoder by identifying a class of high-norm visual tokens from ViT, referred to as *ViT attention sinks* – a problem that has been rarely studied but is indeed very important for LVLMs. Our findings show that these ViT sinks encapsulate high-level semantic concepts from images, allowing the LLM to perform more effective understanding and reasoning. Despite their importance, these sink tokens are often overlooked in existing LVLM architectures. To explore their contribution, we present both qualitative and quantitative analyses of the information embedded in these sink tokens. We also propose both training-free and training-based approaches to better leverage how this information is interpreted by the LLM, and to what extent. By explicitly utilizing these tokens, we demonstrate substantial improvements across a range of LVLMs and visual reasoning tasks, including but not limited to mathematical problem solving, logical inference, and geometric understanding, highlighting the untapped potential of ViT attention sinks in enhancing visual reasoning. [1]

---

*Equal contribution. Listed in alphabetical order.
[1]Project page: DIYSink.

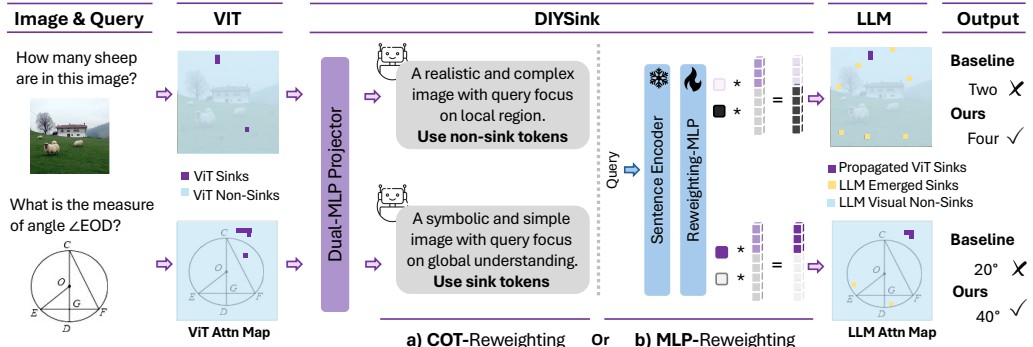

Figure 2: **Overview.** *DIYSink* leverages a Dual-MLP Projector to correctly project sink and non-sink tokens, and one of the two token selection modules, CoT-Reweighting or MLP-Reweighting, to dynamically select the best set of tokens for the LLM based on the specific input.

# 1 INTRODUCTION

Large Vision Language Models (LVLMs) (Liu et al., 2023c; Zhang et al., 2023; Bai et al., 2023; Awadalla et al., 2023; Chen et al., 2023b; Abdin et al., 2024), which integrate the visual capabilities of vision transformers (*e.g.*, ViTs) with the generative capabilities of large language models (LLMs), have demonstrated impressive performance across a broad spectrum of multimodal tasks, spanning visual question answering (Antol et al., 2015; Goyal et al., 2017), mathematical reasoning (Lindström & Abraham, 2022; Lu et al., 2023; Chen et al., 2021), and many others. As these models continue to improve and find increasing deployment, there has been a growing interest in understanding their internal mechanisms, especially their attention dynamics. Attention plays a central role in how LVLMs integrate and align visual and textual inputs. Specifically, attention weights determine how strongly each text (*e.g.*, output) token is influenced by the corresponding (*e.g.*, input) visual token.

One notable emergent behavior in these models is the phenomenon of *attention sinks* (Xiao et al., 2023b; Kang et al., 2025; Sun et al., 2024; Darcet et al., 2024), where the model disproportionately assigns high attention on a small subset of tokens; often irrespective of the input. These tokens often corresponding to non-semantically or uninformative regions such as blank image (visual) or punctuation (language) tokens. This pattern has been consistently observed in ViTs (Darcet et al., 2024; Jiang et al., 2025) and LLMs (Sun et al., 2024; Gu et al., 2024), and naturally extends to LVLMs due to their hybrid architecture.

Attention sinks in LVLM models have generally been shown to be detrimental to model performance, with methodologies centered on identifying and de-emphasizing, if not completely removing, them in inference. Specifically, Kang et al. (2025) shows that masking visual sinks leads to little effect and redistributing attention to other tokens actually leads to improvement; Darcet et al. (2024) prepend register tokens to absorb high-norm tokens in ViT and achieve better unsupervised object discovery with the register-free feature maps. These findings, however, are seemingly in contrast to recent findings in LLM models that find that attention sinks are implicitly useful and encode indispensable biases (Sun et al., 2024), which could be beneficial for long-context handling and to reduce over-mixing (Barbero et al., 2025). This raises an important question – *Are there any fundamental benefits to sink tokens in LVLM models, and if so, can they be understood and operationalized?*

To address this question, we first carefully study the emergence of sink tokens in LVLMs, showing that they result in a combination of sink tokens from ViT backbone being propagated to LVLM (Fig. 2 (purple)) and sink tokens from the LLM itself (Fig. 2 (yellow)) – an observation not previously made. We then focus on analyzing, the less explored, ViT sinks (Fig. 2 (purple)) – making three core findings: (1) The propagated ViT sinks capture coarse, high-level contextual information, (2) these tokens benefit certain tasks that require high-level image understanding or reasoning, and (3) given the seemingly different level of semantics encoded in these sinks vs. other visual tokens (global vs. local), using a single learned projection across both diminishes their effectiveness in practice.

Armed with these findings, we propose an effective way to dynamically enhance LVLM performance, by emphasizing and (sometimes) de-emphasizing ViT sinks depending on the task and image content.

Specifically, first, leveraging (1) and taking advantage of the LLM's causal structure, we propose a straightforward training-free approach on enhancing LVLM performance by moving the ViT sink tokens to the front. This shows improvements across a variety of LVLM models, and particularly for tasks that require a high-level understanding or reasoning. Second, we introduce *DIYSink*, illustrated in Fig. 2 (middle), a training-based framework designed to help LVLMs make better use of ViT sink tokens. The framework relies on *dual MLP projection* trained separately to most effectively use ViT sink and non-sink tokens by the LLM – addressing (3). Given the dual MLP projection LVLM variants, we then leverage two mechanisms to dynamically select which class of tokens should be used in inference (sink, non-sink or both) and to what extent; operationalization observation (2). One mechanism leverages hard selection obtained through Chain-of-Though (CoT) routing, while another trains a lightweight soft weighting module with a small amount of cross-task data.

We validate proposed *DIYSink* using four ViT–LLM combinations, with SigLIP (Zhai et al., 2023)and CLIP-VIT (Radford et al., 2021) as vision backbones and Qwen2 (Yang et al., 2024), Qwen2.5 (Qwen et al., 2025), Phi-2 (Abdin et al., 2024), and Vicuna (Zheng et al., 2023) as language backbones of varying sizes. Our approach consistently improves performance across a wide range of benchmarks.

**Contributions:** To summarize, our contributions are two fold. On one hand, they take the form of analysis that sheds light on the role and function of ViT sinks within LVLM models. On the other hand, it leverages those observations, to propose simple yet systematic and effective improvements for both open-source (training-based) and closed-source (training-free) variants of these models.

## 2 PRELIMINARIES

Large Vision Language Models (LVLMs) comprise of three key components: (1) *Vision Encoder* $\mathcal{E}_v$ (*e.g.*, CLIP, SigLIP): extracts visual features from input images, producing hidden state $\mathbf{V} = [\mathbf{v}_1, \mathbf{v}_2, \ldots, \mathbf{v}_n], \mathbf{v}_i \in \mathbb{R}^{D'}$, where $D'$ is the hidden dimension of ViT and $n$ is a number of image patches. (2) *Connector* module $\mathcal{E}_c$ (*e.g.*, Q-Formers (Li et al., 2023), MLP Projectors (Liu et al., 2023c)): maps the visual features to the textual space of the LLM, producing visual tokens $\mathcal{I}_{\text{vis}}$. (3) *Language Model* $\mathcal{E}_l$ (*e.g.*, LLaMA (Touvron et al., 2023), Qwen (Bai et al., 2023), Phi (Abdin et al., 2024)): an autoregressive transformer that takes in a sequence of tokens, namely, system tokens ($\mathcal{I}_{\text{sys}}$), visual tokens ($\mathcal{I}_{\text{vis}}$), and query tokens ($\mathcal{I}_{\text{txt}}$), and processes them layer by layer to generate outputs ($\mathcal{I}_{\text{out}}$). Each token has a hidden state $\mathbf{x}_i^l \in \mathbb{R}^D$ at layer $l$, which is updated through attention and feedforward networks. During multi-head attention (MHA), each token can attend to others. Notably, MHA in ViTs is non-causal, allowing full token interaction, while MHA in LLMs is causal, restricting attention to past tokens only. Following the definition in Kang et al. (2025), we define the attention weights $\alpha_{i,j}^{l,h}$ to be the strength with which token $i$ attends to $j$ at layer $l$ and head $h$. We focus specifically on visual attention, analyzing interactions between $\mathcal{I}_{\text{vis}}$ and the text: $\mathcal{I}_{\text{txt}}$ and $\mathcal{I}_{\text{out}}$.

**Attention sinks and massive activation.** Sun et al. (2024); Kang et al. (2025); Darcet et al. (2024); Cancedda (2024); Gu et al. (2024); Yu et al. (2024); Zhang et al. (2024); Ge et al. (2023); Xiao et al. (2023a); Barbero et al. (2025) observed that sink tokens are common in transformer-based models including ViTs and LLMs. These sink tokens, usually receive disproportionately high attention, tend to exhibit abnormally high values (so called massive activation) in particular hidden dimensions, referred to as sink dimensions $D_{\text{sink}}$ (Kang et al., 2025), and have high feature norms. As such, sink tokens can be identified by the following definition:

$$\hat{\mathcal{I}}^l = \left\{ j \in \mathcal{I} \,\middle|\, \phi\left(\mathbf{x}_j^{l-1}\right) \geq \tau \right\}, \tag{1}$$

where $\phi$ is a *sink characteristic function*, and $\tau$ is the pre-defined threshold. The *sink characteristic function* $\phi$ can be: (1) feature norms of the token (Darcet et al., 2024): $\phi\left(\mathbf{x}_j^{l-1}\right) = \|\mathbf{x}_j^{l-1}\|$, (2) average attention weights on the token : $\phi\left(\mathbf{x}_j^{l-1}\right) = \frac{1}{|\mathcal{I}_{\text{out}}|} \sum_{i \in \mathcal{I}_{\text{out}}} \alpha_{i,j}^{l,h}$, where $h$ denotes the head of interest, and (3) sink dimension value (Sun et al., 2024) of the token : $\phi(\mathbf{x}_j^{l-1}) = \max_{\check{d} \in \mathcal{D}_{\text{sink}}} \left|\text{RMSNorm}(\mathbf{x}_j^{l-1})[\check{d}]\right|$ , where $\mathcal{D}_{\text{sink}}$ is the pre-defined dimensions (*e.g.*, $\mathcal{D}_{\text{sink}} = \{1415, 2533\}$ for LLaMA2-7B), and RMSNorm is Root Mean Square Normalization.

In this paper, we focus on *ViT sinks* in ViT denoted as $\hat{\mathcal{I}}_{\text{vit}}^l$, *propagated ViT sinks* from ViT to LLMs as $\hat{\mathcal{I}}_{\text{vit}\rightarrow\text{llm}}^l$, and the *LLM-emerged sinks* $\hat{\mathcal{I}}_{\text{llm}}^l$. Following Darcet et al. (2024), we identify $\hat{\mathcal{I}}_{\text{vit}}^l$ with the sink characteristic function $\phi$ as a feature norm in Eq. (1) and $\tau = 100$ (details in Sec. B.1). For the LLM sinks $\hat{\mathcal{I}}_{\text{llm}}^l$, we follow Kang et al. (2025), defining $\phi$ as the sink dimension value with $\tau = 20$.

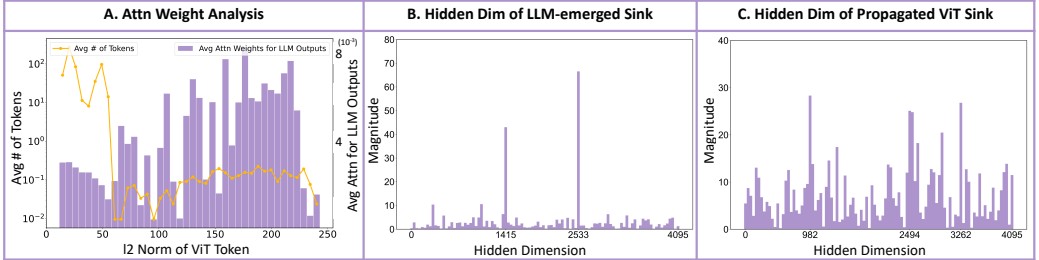

Figure 3: **Attention to ViT sink tokens and sink dimensions of ViT and LLM sinks in LLaVA-v1.5-7B.** (A) compares ViT token norms with the attention assigned during LLM decoding. (B)–(C) show the sink-dimension distributions for LLM-emergent sinks and ViT-propagated sinks. For enlarged version, please refer to Fig. A10.

## 3 ViT Sinks in LVLMs

We now provide our analysis on ViT sink tokens within the LVLMs, examining how they propagate through the model, what information they encode, and how they affect the behavior of the LLM. We show analysis on LLaVA-7B, and also provide the analysis results on other LVLMs in Sec. A. The observations are also consistent across vision encoders (*i.e.*, CLIP, SigLIP).

### 3.1 The Property of Propagated ViT sinks in LLMs

**Propagation of ViT sink tokens into LLM.** ViT sinks, as studied in Darcet et al. (2024), exhibit high vector norms. Given that many related works use attention weight to identify the sink tokens in LLM (Sun et al., 2024; Kang et al., 2025; Barbero et al., 2025; Yu et al., 2024), to quantitatively verify whether these high-norm vectors are being propagated into the LLM as sinks, we visualize the relationship between ViT token norms calculated in ViT and the attention weights they receive from the LLM during output generation, with results averaged across 300 image-question pairs. As shown in Fig. 3(A), the x-axis here represent the categorized token norm. The left y-axis and the yellow line represent the average number of tokens fall in each bin. The right y-axis and the purple bars represent the attention weight assigned from the LLM to the visual tokens in that bin during output generation. We observe a positive correlation between ViT token norm and the attention weight assigned by the LLM during generation. From this, we conclude:

*Tokens with higher norms in the ViT are more likely to receive higher attention weights and become sinks in the LLM.*

Specifically, we find that most tokens have a norm below 60 in the ViT. Only a small number of tokens (typically 3–5 per image) have norms above 100, yet these high-norm tokens receive substantially higher attention weights, approximately 7 times greater than those assigned to the rest of the tokens. We highlight that this correlation is not architecturally enforced, making it a non-trivial finding. It suggests that the LLM implicitly inherits ViT's internal saliency signals, revealing a strong inductive link between visual and language components in LVLMs.

**Hidden dimension breakdown of propagated ViT sink in LLM.** We further investigate how visual sink tokens from the ViT propagate to LLM. In Fig. 3(B), we plot the hidden dimension magnitude of the LLM-emerged sinks $\hat{\mathcal{I}}_{\text{llm}}$ and, in Fig. 3(C), the hidden dimension magnitude of propagated ViT sinks $\mathcal{I}_{\text{vit}\rightarrow\text{llm}}$. The values are from the second last layer of the LLM, averaged over 300 examples. Across these examples, we observe that ViT sinks have high activation consistently on certain hidden dimensions (*i.e.*, 982, 2494, and 3263) within the LLM, regardless of the input image or prompt. These dimensions are different from the sink dimensions of LLM sinks. In summary:

*Visual sink tokens propagate into the LLM as distinct sink tokens, activating different hidden dimensions of the sinks emerged in LLM.*

Moreover, these high value sink dimensions emerge only after multimodal training. In LLaVA-7B, the LLM's original sink dimensions are {2533, 1415}, while the propagated ViT sink tokens activate dimensions {982, 2494, 3263}. This distinction is critical: "Prior studies (Kang et al., 2025) identify

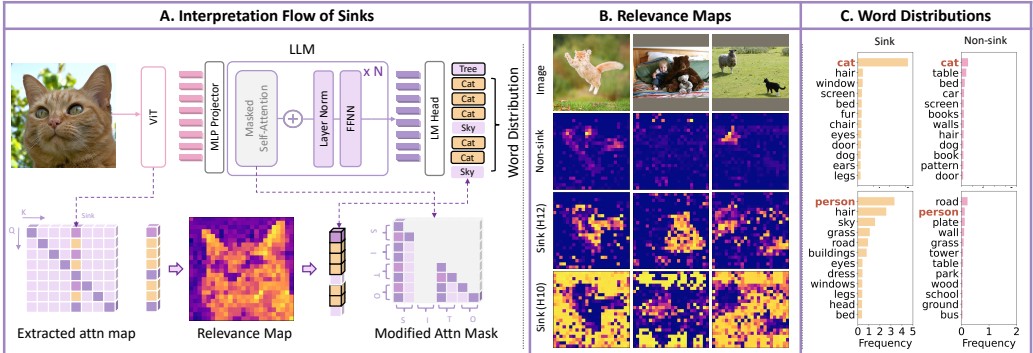

Figure 4: (A) Flow of obtaining relevancy map and word distribution. (B) Relevance map of sink and non-sink tokens. H12 and H10 denote the respective foreground and background attention head of the penultimate layer of the LLM used to extract the relevance maps. (C) Word Distribution of sink and non-sink tokens from the foreground head (H12). The first and second rows represent the word distributions obtained from 300 images where cat and person are the main objects.

sinks tokens according to LLM original sinks dimensions (Sun et al., 2024) and found redistributing their attention improve overall performance. This may inadvenrtently conflate the effect of two different sinks." We argue that explicitly disentangling them is essential to understanding their distinct behaviors, given the fundamentally different architectures they originate from.

**Impact of ViT sink tokens in LLM.** With the aforementioned observations, we are able to identify the ViT-propagated sink tokens within the LLM. Lastly, we investigate their importance by analyzing the average attention weights received by ViT sink tokens. By computing the average attention weights from output tokens to target tokens across $1,000$ image-question pairs, we find that, on average, non-sink tokens receive $0.1532\%$ attention per token, LLM-emerged sink tokens receive $1.27\%$, and ViT sink tokens receive $1.13\%$. This illustrates the importance of ViT sink tokens.

## 3.2 WHAT IS IN THE ViT SINKS AND HOW DOES LLM INTERPRET THEM?

As shown in the previous section, ViT sinks have a significant impact on the model's output, suggesting they might encode information critical for model learning and inference. To better understand the content and role of these tokens, we examine the attention mechanism in both the ViT and the LLM.

**Interpreting sink tokens via *Relevance Map* in the attention layer.** In transformer-based models like ViT, attention maps provide insights into how the model aggregates information (Kovaleva et al., 2019; Reif et al., 2019). In these maps, vertical columns indicate how much attention a particular token receives from all other tokens, reflecting the relevance or importance of the target token during processing. As illustrated in Fig. 4 (A), we visualize the vertical column corresponding to the sink in the given attention map. After reshaping and normalizing this attention column, we obtain a 2D map with the same spatial layout as the image patches, referred to as *Relevance Map*. Fig. 4 (B) presents relevance maps for sink and non-sink tokens across three images. Note that the attention maps are extracted from the second-to-last layer, with head 10 (H10) used for background and head 12 (H12) for foreground in CLIP-ViT. We observe that the non-sink token shows high relevance primarily to its local neighbors, whereas the sink token receives attention from tokens broadly distributed across either foreground or background regions, indicating that –

*ViT sinks capture coarse-grained, high-level contextual features aligned with the specific focus of each attention head.*

**Decoding ViT sink tokens into word distributions.** Using the relevance map, we qualitatively interpret the information encoded in ViT sink tokens. To analyze it quantitatively, we decode visual tokens into word distributions leveraging the LLM. Together with the relevance map, we can analyze multiple images and collect word distribution associated with the target token. Inspired by previous works of ViT concept discovery (Rao et al., 2024; Chen et al., 2023a), we disable attention from all tokens to visual tokens in the LLM, as illustrated by the modified attention mask in Fig. 4 (A), to

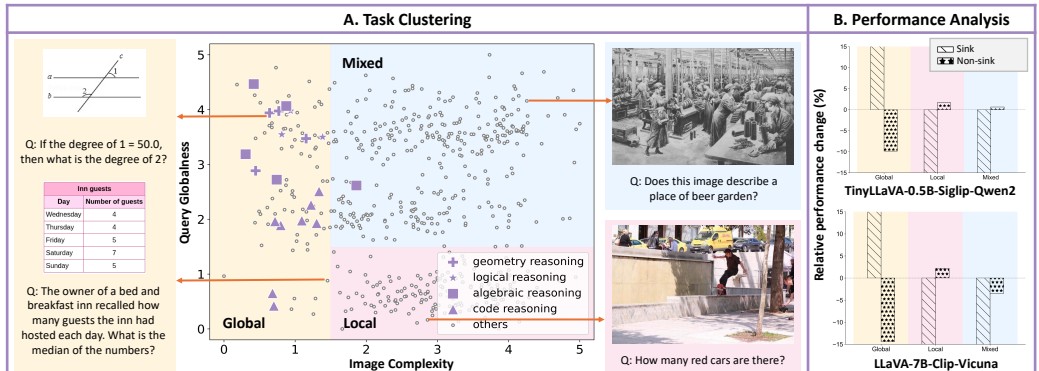

Figure 5: (A) Task Clustering based on GPT-4o annotated Image Complexity and Query Globalness. (B) Performance analysis of two model variants (*i.e. Sink-only* and *Non-sink-only*) for evaluating the influence of ViT sink tokens on different tasks.

prevent information exchange. By forwarding these isolated visual tokens through all layers, we are able to map their embeddings to the output vocabulary, generating word predictions for each.

We visualize the word distributions for 300 cat images and 300 person images, as shown in Fig. 4 (C). We observe that sink tokens are strongly associated with the main object (*i.e.*, "cat" or "person"), whereas non-sink tokens yield far fewer semantically aligned terms. These quantitative results across multiple images support our finding that ViT sink tokens encode coarse-grained, high-level contextual features and are semantically meaningful.

## 3.3 TASKS CLASSIFICATION AND PRELIMINARY EXPERIMENT

**Interpretation and hypothesis.** Above observations that ViT sink tokens are significant and appear to carry course-grained high-level contextual information lead to a hypothesis regarding their use. Specifically, (1) we hypothesis that such compact high-level contextual information could be very useful for tasks that require it (*e.g.*, scene recognition) and may distract the model when the task is highly local and unlikely to be captured in such high-level context (*e.g.*, localization). Further, (2) given the fixed capacity of these tokens, benefit of them can be reduced for images that are complex and cannot be summarized effectively. To validate this hypothesis, we setup a simple experiment.

**Task taxonomy via query and image properties.** To validate our hypothesis, we curated a dataset of 600 image-query pairs, uniformly sampled from a suite of widely used benchmarks, including GQA (Hudson & Manning, 2019), TextVQA (Singh et al., 2019b), ScienceQA (Lu et al., 2022), MME (Fu et al., 2023), MathVista (Lu et al., 2023), Each instance was annotated using GPT-4o with two ratings: (i) *image complexity*, measuring the visual density and richness of the scene, and (ii) *query globalness*, assessing whether the question depends on high-level contextual reasoning or fine-grained spatial cues. Based on these continuous annotations (normalized to [0, 5]), we categorize each instance into one of three clusters (see Fig. 5 (A)): **Global tasks** (low image complexity), **Local tasks** (high complexity and low query globalness), and **Mixed tasks** (the rest).

**Effect of ViT sink tokens on downstream tasks.** We investigate the influence of ViT sink tokens during inference using two configurations: (1) *Sink-only* and (2) *Non-sink-only*, where we only use ViT sinks or non-sinks as visual input to the LLM for inference. From the results shown in Fig. 5 (B), we observe that *Sink-only* configurations yield strong performance specifically on global tasks, suggesting the compact high-level contextual information in ViT sink can be very useful for tasks that require it. In contrast, for local tasks, removing sinks improves performance, indicating potential interference. This observation confirms that

*ViT sink tokens encode useful semantic summaries, but only useful under the right conditions. They are beneficial for tasks with low visual complexity and global semantics but may degrade performance on tasks that demand localized, detail-rich visual processing.*

This context-dependent behavior underscores the importance of adaptive sink usage and motivates further architectural design around task-aware visual token selection. For more analysis results,

Table 1: **Performance impact of reordering ViT sinks to the front.** Green and red indicate increase or decrease over the corresponding baseline. Subscripts show the absolute change from baseline.

| Methods | LLaVA eval | | | | | MME | | | | | MathVista | | | |
|---|---|---|---|---|---|---|---|---|---|---|---|---|---|---|
| | AVG | GQA | SQA | TextVQA | MMMU | ALL | Cognition | ComRea | TextTrans | CodeRea | ALL | ALG | GEO | LOG |
| InternVL2.5-4B | 70.99 | 62.11 | 96.18 | 73.47 | 50.20 | 2332.16 | 645.35 | 142.86 | 200.00 | 177.50 | 62.90 | 65.48 | 66.53 | 16.22 |
| + Sink-to-the-front | 71.01 $_{+0.02}$ | 62.39 | 96.06 | 73.17 | 50.40 | 2351.33 $_{+19.17}$ | 645.35 | 145.00 | 200.00 | 175.00 | 63.30 $_{+0.40}$ | 65.12 | 68.20 | 16.22 |
| Phi 3.5 V | 65.29 | 63.48 | 91.25 | 65.24 | 43.20 | 1887.90 | 412.14 | 137.14 | 87.50 | 97.50 | 43.10 | 40.21 | 38.49 | 10.81 |
| + Sink-to-the-front | 65.71 $_{+0.42}$ | 63.49 | 91.31 | 64.05 | 44.00 | 1891.27 $_{+3.37}$ | 433.92 | 136.42 | 87.50 | 112.50 | 43.50 $_{+0.40}$ | 40.57 | 38.49 | 13.51 |
| Deepseek-vl-7b-chat | 61.35 | 61.23 | 81.85 | 64.20 | 38.10 | 1787.34 | 299.28 | 129.28 | 85.00 | 35.00 | 37.20 | 30.96 | 32.22 | 16.22 |
| + Sink-to-the-front | 61.56 $_{+0.21}$ | 61.31 | 81.99 | 64.34 | 38.6 | 1786.40 $_{-0.94}$ | 301.42 | 131.42 | 85.00 | 32.50 | 37.70 $_{+0.50}$ | 31.32 | 32.22 | 16.22 |
| Molmo-7B-D | 61.68 | 54.83 | 86.30 | 61.49 | 44.10 | 1821.93 | 372.14 | 122.14 | 72.50 | 87.50 | 49.40 | 37.01 | 35.56 | 8.11 |
| + Sink-to-the-front | 61.74 $_{+0.06}$ | 54.96 | 86.35 | 60.65 | 45.00 | 1852.21 $_{+30.28}$ | 379.64 | 127.14 | 72.50 | 87.50 | 51.20 $_{+1.80}$ | 37.37 | 35.98 | 13.51 |
| Gemma3-12B-it | 61.26 | 45.24 | 82.55 | 70.23 | 47.00 | 1706.30 | 535.36 | 127.86 | 112.50 | 162.50 | 40.80 | 48.40 | 48.95 | 13.51 |
| + Sink-to-the-front | 61.55 $_{+0.29}$ | 45.41 | 82.50 | 70.30 | 48.00 | 1740.11 $_{+33.81}$ | 563.21 | 130.71 | 130.00 | 170.00 | 41.00 $_{+0.20}$ | 50.89 | 51.46 | 8.11 |
| Meteor | 65.71 | 62.54 | 94.29 | 67.01 | 39.00 | 2188.75 | 508.21 | 140.71 | 160.00 | 102.50 | 52.80 | 40.21 | 42.68 | 13.51 |
| + Sink-to-the-front | 65.85 $_{+0.14}$ | 62.57 | 94.65 | 67.07 | 39.10 | 2198.47 $_{+9.72}$ | 516.42 | 141.42 | 160.00 | 110.00 | 53.00 $_{+0.20}$ | 40.21 | 42.68 | 13.51 |

including ViT sink tokens behavior across different checkpoints and linear-probing test of the semantics of VIT sink tokens, please refer to the Sec. A.3 and Sec. A.4.

## 4 LARGE VISION LANGUAGE MODEL (LVLM) FRAMEWORK REDESIGN

### 4.1 TRAINING-FREE APPROACH: FOR WHEN TRAINING IS NOT AVAILABLE.

Motivated by the analysis in Sec. 3.3, we propose a simple yet effective inference-time strategy called *sink-to-the-front*, where ViT sink tokens are repositioned to the beginning of the visual token sequence. This allows subsequent tokens to refer to sink tokens and can benefit mixed and global tasks that require high-level context (see left side of Fig. 5), while maintaining performance on local tasks. This approach requires no additional training and can be applied post hoc to any existing LVLM. Specifically, in the inference phase, given a visual token sequence from the vision encoder in a LVLM, we first identify the ViT sinks by the definition in Eq. (1), where the token characteristic is measured by its feature norm. We jointly reposition them and their corresponding positional embeddings at the beginning of the visual token sequence before passing the sequence to the connector and LLM.

### 4.2 TRAIN-FROM-SCRATCH: IMPROVING INFORMATION FLOW OF MODELS

The observation in the analysis (Sec. 3.3) suggests that ViT sinks are useful semantic summaries, but only useful under the right conditions. To fully and dynamically leverage the "power" of sink, we propose a train-from-scratch approach, *DIYSink*, to improve the visual information flow in LVLM.

*DIYSink* introduces two key design components: (1) *Dual-MLP Projection Layers*, which independently process sink and non-sink ViT tokens, preventing two representations from being conflated; and (2) *Dynamic Token Selection Modules*, which leverage inputs as a gating mechanism to help the LVLM decide which set of visual tokens (*i.e.*, ViT sinks, non-sinks, or both) to use during inference.

**Dual-MLP Projection Layers.** As discussed in Sec. 3.1, sink tokens exhibit unique characteristics such as high activation and large norms that differ significantly from those of non-sink tokens. As a result, it is challenging for a shared MLP connector to effectively project both types of tokens into a common semantic space that aligns with the LLM's expectations. To address the issue, in *DIYSink*, we introduce a dual-MLP projector. Each MLP is trained exclusively on either sink or non-sink tokens, enabling it to specialize in projecting its token type into an LLM-compatible embedding space. Formally, let $\mathbf{V}_{\text{sink}}$ and $\mathbf{V}_{\text{non-sink}}$ be two disjoint sets of ViT sink and non-sink visual tokens, respectively. We define two separate MLP connectors: $f_{\text{sink}} : \mathbb{R}^{D'} \to \mathbb{R}^{D}$ for projecting sink, and $f_{\text{non-sink}} : \mathbb{R}^{D'} \to \mathbb{R}^{D}$ for projecting non-sink tokens.

During pretraining, each connector is optimized independently using only its corresponding token:

$$\min_{\theta_{f_{\text{sink}}}} \mathcal{L}_{\text{LM}}(\mathcal{E}(\mathcal{I}_{\text{sys}}, f_{\text{sink}}(\mathbf{V}_{\text{sink}}), \mathcal{I}_{\text{txt}}, \mathcal{I}_{\text{out}})), \quad \min_{\theta_{f_{\text{non-sink}}}} \mathcal{L}_{\text{LM}}(\mathcal{E}(\mathcal{I}_{\text{sys}}, f_{\text{non-sink}}(\mathbf{V}_{\text{non-sink}}), \mathcal{I}_{\text{txt}}, \mathcal{I}_{\text{out}})), \quad (2)$$

Table 2: **Full benchmark comparison.** ReW and CoT denote our reweighting module and Chain-of-Thought prompting, respectively. Green indicates increase over the corresponding baseline. † denotes models are fine-tuned on original LLaVA fine-tuning datasets Liu et al. (2023c).

| Methods | LLaVA eval | | | | | MME | | | | | MathVista | | | |
|---|---|---|---|---|---|---|---|---|---|---|---|---|---|---|
| | AVG | GQA | SQA | TextVQA | MMMU | All | Cog. | ComRea | TextTrans | CodeRea | All | ALG | GEO | LOG |
| **TinyLLaVA-0.5B-SigLIP-Qwen2-0.5B** | | | | | | | | | | | | | | |
| Baseline[†] | 48.34 | 57.98 | 57.76 | 47.32 | 30.30 | 1381.10 | 207.14 | 82.14 | 50.00 | 37.50 | 24.30 | 22.42 | 20.92 | 24.32 |
| *DIYSink* (CoT) | 49.04 $_{+0.70}$ | **58.14** | **61.08** | 46.14 | 30.80 | 1456.78 $_{+75.68}$ | **277.50** | 85.00 | **80.00** | **52.50** | 25.10 $_{+0.80}$ | 24.91 | **23.85** | 24.32 |
| *DIYSink* (ReW) | 49.13 $_{+0.79}$ | 57.75 | 60.24 | **47.42** | **31.10** | 1451.87 $_{+70.77}$ | 229.64 | **87.14** | 50.00 | 37.50 | 25.20 $_{+0.90}$ | **25.27** | 23.01 | 24.32 |
| **TinyLLaVA-3.1B-SigLIP-Phi2-3B** | | | | | | | | | | | | | | |
| Baseline[†] | 48.55 | 51.09 | 68.47 | 41.15 | 33.50 | 1455.22 | 261.79 | 99.29 | 50.00 | **57.50** | 25.90 | 23.49 | 19.67 | 18.92 |
| *DIYSink* (CoT) | 52.17 $_{+3.62}$ | **59.99** | 68.57 | 46.22 | 33.90 | 1523.18 $_{+67.96}$ | **280.71** | 110.71 | **67.50** | 50.00 | 26.20 $_{+0.30}$ | **29.54** | **27.20** | 18.92 |
| *DIYSink* (ReW) | 54.34 $_{+5.79}$ | 59.79 | **70.50** | **50.77** | **36.30** | 1682.41 $_{+227.19}$ | 265.71 | **115.71** | 57.50 | 45.00 | 27.40 $_{+1.50}$ | 28.11 | 26.78 | 16.22 |
| **TinyLLaVA-3.1B-SigLIP-Qwen2.5-3B** | | | | | | | | | | | | | | |
| Baseline[†] | 70.22 | 62.47 | 74.12 | 58.35 | **40.20** | 1740.01 | 293.21 | 115.71 | 65.00 | **52.50** | 30.40 | 37.01 | 38.08 | 18.92 |
| *DIYSink* (CoT) | 70.98 $_{+0.76}$ | **63.62** | 75.41 | 58.54 | 39.60 | 1758.47 $_{+18.46}$ | **302.14** | 127.14 | 70.00 | 47.50 | **33.40** $_{+3.00}$ | **41.64** | 42.68 | **24.32** |
| *DIYSink* (ReW) | 71.65 $_{+1.43}$ | 63.58 | **76.20** | **60.72** | 39.00 | 1761.41 $_{+21.40}$ | 299.28 | **129.28** | 70.00 | 47.50 | 33.10 $_{+2.70}$ | 41.63 | **43.10** | 24.32 |
| **LLaVA-7B-CLIP-ViT-Vicuna-7B** | | | | | | | | | | | | | | |
| Baseline[†] | 56.55 | **62.65** | 68.91 | 58.82 | 35.80 | 1781.70 | 282.50 | **130.00** | 55.00 | 42.50 | 26.00 | **25.27** | 20.92 | 13.51 |
| *DIYSink* (CoT) | 56.76 $_{+0.21}$ | 62.57 | 69.56 | **58.99** | **35.90** | 1797.80 $_{+10.10}$ | 310.36 | 127.86 | 77.50 | **57.50** | 26.40 $_{+0.40}$ | 24.56 | 21.76 | **13.51** |
| *DIYSink* (ReW) | 56.59 $_{+0.04}$ | 62.51 | **70.00** | 58.16 | 35.70 | 1787.00 $_{+5.30}$ | **332.85** | 127.85 | **95.00** | 57.50 | 26.60 $_{+0.60}$ | 22.78 | 18.83 | 13.51 |

where $\mathcal{L}_{LM}$ is the language modeling loss used for pretraining. With the separately-trained MLPs, in the fine-tuning phase, we use them to convert ViT sinks and non-sinks independently, concatenate all sinks with all non-sink tokens, and follow the standard LLM fine-tuning procedure (Liu et al., 2023c).

**Dynamic token selection.** Given the dual MLP projection, we explore two mechanisms to dynamically select which class of tokens should be used in inference (sink, non-sink, or both) based on input complexity and task demands. One mechanism leverages hard selection obtained through Chain-of-Thought (CoT) routing, while the other utilizes a lightweight soft weighting module trained from a small amount of cross-task data.

In Sec. 3.3, we empirically observed that sink tokens are effective for scene-level, holistic understanding, whereas non-sink tokens excel at capturing fine-grained details. Based on this observation, we design a two-step chain-of-thought process to pre-classify the given task: (1) determine whether the image is symbolic (with minimal local detail) or a real-world photographic scene, and (2) assess whether the question query requires holistic reasoning or local visual understanding. Following these criteria, if the task involves a symbolic/simple image and holistic reasoning, we use only ViT sink tokens for inference. If the task involves a real-world/complex image and local visual understanding, we use only non-sink tokens. For all other mixed or ambiguous cases, we leverage both token types to generate the final answer. We note that this is equivalent to hard [0/1] token selection.

In addition to the chain-of-thought method, we also explore a learnable reweighting mechanism to dynamically balance the contribution of sink and non-sink tokens before feeding them into the LLM, as illustrated in the Reweighting MLP section of Fig. 2.

Given an input text question embedding $\mathbf{q} \in \mathbb{R}^d$ obtained from sending input query into a frozen sentence encoder, where $d$ represent the hidden dimension of the encoded sentence feature, the reweighting MLP $\mathcal{R}$ outputs two scalar weights: $[w_{sink}, w_{non\text{-}sink}] = \mathcal{R}(\mathbf{q}) \in \mathbb{R}^2$. We then use the output weights to reweight the sink and non-sink tokens and input their concatenation into the LLM:

$$\min_{\theta_\mathcal{R}} \mathcal{L}_{LM}(\mathcal{E}(\mathcal{I}_{sys}, \underbrace{[\{w_{sink} \cdot f_{sink}(\mathbf{V}_{sink})\}; \{w_{non\text{-}sink} \cdot f_{non\text{-}sink}(\mathbf{V}_{non\text{-}sink})\}]}_{\mathcal{I}_{vis}}, \mathcal{I}_{txt}, \mathcal{I}_{out})) \quad (3)$$

During training, only the parameters of $\mathcal{R}$ are updated, while all other components remain frozen to prevent additional information leakage and maintain the fairness of evaluation.

## 5 EXPERIMENTS

**Baselines and training.** We evaluate our method on four LVLM (ViT–LLM) configurations. For the vision backbone we employ SigLIP (Zhai et al., 2023) and CLIP-ViT-L (Radford et al., 2021),

while for the language backbone we use Qwen2 (Yang et al., 2024), Qwen2.5 (Qwen et al., 2025), Phi-2 (Abdin et al., 2024), and Vicuna (Zheng et al., 2023) spanning various scales (0.5B, 3B, 7B). All models are pre-trained and fine-tuned on official LLaVA training sets (Liu et al., 2023c). For reweighting MLP, training samples are drawn from the training sets of PixMo (Deitke et al., 2024) and GeoQA (Chen et al., 2021), which are distinct from our evaluation benchmarks. We use 120 examples for 0.5B and 3B models, 240 examples for the 7B model (see Sec. B.4 for details).

**Evaluation Benchmarks.** We conduct a wide range of evaluations on various benchmarks including general benchmarks used in LLaVA (or LLaVA eval) (Liu et al., 2023c), GQA (Hudson & Manning, 2019), ScienceQA (Lu et al., 2022), TextVQA (Singh et al., 2019a), MMMU (Yue et al., 2024), fine-grained benchmark like MME (Fu et al., 2023) and mathematical reasoning benchmark MathVista (Lu et al., 2024b) to examine the influence of *DIYSink* on LVLMs following the common protocols of these benchmarks. We provide benchmark details in Sec. B.5

**Training-free method.** We evaluate the proposed approach on several recent LVLMs, including InternVL2.5 (Zhu et al., 2025), Phi-3.5 (Abdin et al., 2024), DeepSeek-VL (Lu et al., 2024a), Molmo (Deitke et al., 2024), Gemma3 Team et al. (2025) and Meteor Lee et al. (2024). As shown in Tab. 1, our lightweight method consistently improves performance on benchmarks require holistic understanding of the image, particularly MathVista (*e.g.*, 1.8 for Molmo). We also observe modest yet consistent improvements on benchmark with mixed tasks (i.e., MME). Interestingly, we also observe improvements on Meteor Lee et al. (2024), a VLM built on a Mamba-based language model, indicating that the benefits of sink-to-the-front extend beyond Transformer-based architectures. We additionally note that DeepSeek-VL's channel-wise concatenation of SAM and SigLIP features creates a blended features, complicating the token behavior; consequently, the gains are smaller.

**Training-from-scratch method.** We evaluate our proposed modules across multiple benchmarks as shown in Tab. 2. Our results show that with correct projection and reweighting of sink tokens, the model consistently improves on global reasoning tasks such as MME code reasoning and MathVista's algebraic, geometric, and logical reasoning. For instance, applying *DIYSink* (CoT) or *DIYSink*(ReW) to the LLaVA-7B (Liu et al., 2023c) both yield a 15-point gain on MME code reasoning. On MathVista, *DIYSink* (CoT) delivers notable gains of 6.05 and 7.53 points in algebra and geometric reasoning, respectively, when applied to TinyLLaVA-Phi2-3B and gains of 4.62 and 5.02 when applied *DIYSink* (ReW) to TinyLLaVA-Qwen2.5-3B. In addition to domain-specific reasoning improvements, we also observe consistent gains on commonly evaluated LLaVA benchmarks. For instance, *DIYSink* (CoT) and *DIYSink* (ReW) improve the LLaVA-eval average by 3.62 and 5.79 points, respectively, over TinyLLaVA-3B. These results suggest that our approach effectively mitigates representational interference (see Sec. 3.3) and enhances performance across tasks.

Overall, we observe that both explored strategies, CoT and ReW, are effective. The CoT is simpler, but requires additional computation in inference. ReW on the other hand has negligible compute overhead, but does require additional training. Each strategy may be beneficial depending on circumstances.

**Model Ablation.** We conduct an ablation study to evaluate the contribution of the dual-MLP design, which facilitates the conversion of both ViT sink and non-sink information into a format interpretable by the LLM. As shown in Tab. 3, our learned dual-MLP model, even without the token selection module, out-perform baseline across benchmarks. These improvements support our hypothesis that the shared connector design commonly used in LVLMs struggles to effectively convert both ViT sink and non-sink tokens. Due to page limit, we place additional studies in the appendix, includ-

Table 3: **Ablation of Dual-MLP projector.**

| Method | SQA | MMMU | MME ALL | MME Cog. |
|---|---|---|---|---|
| **TinyLLaVA-0.5B (Qwen)** | | | | |
| Baseline | 57.76 | 30.30 | 1381.10 | 207.14 |
| + Dual-MLP (our) | **60.63** | **30.80** | **1439.21** | **208.21** |
| **TinyLLaVA-3B (Phi)** | | | | |
| Baseline | 68.47 | 33.50 | 1455.22 | 261.79 |
| + Dual-MLP (our) | **70.75** | **34.80** | **1679.24** | **262.14** |

ing justification of the design choice of placing sink tokens to the front rather than the end of the sequence (Sec. C.1), ablation on different connector types (Sec. C.3), computation cost (Sec. C.6), the human study on the task taxonomy classifier using GPT(Sec. C.11), an analysis of the learned weights in the selection module (Sec. C.4), comparison on compositional scenes (Sec. C.5), and qualitative examples (Sec. D).

## 6    RELATED WORK

**Large Vision Language Model.** LVLMs (Bai et al., 2025; Team et al., 2023; Achiam et al., 2023; Lu et al., 2024a) typically consist of three key components: (i) a Vision Transformer (ViT), (ii) a connector module, and a (iii) large language model (LLM). Among open-source LVLMs, widely adopted vision backbones include the CLIP-ViT family (Liu et al., 2023b; Abdin et al., 2024), SigLIP (Chen et al., 2023c), and InternViT (Zhu et al., 2025), each selected for its strong pretraining and transfer capabilities. To bridge vision and language, most models utilize either a lightweight multi-layer perceptron (MLP) (Chen et al., 2024; Zhou et al., 2024; Deitke et al., 2024) or cross-attention mechanisms (Zhu et al., 2023; Li et al., 2023). In this work, we focus on the visual encoding component and evaluate our proposed method, both in training-free and training-based settings, across a range of LVLMs that span all three major ViT families and include seven distinct LLMs.

**Attention sinks.** Attention sinks have been observed in well-trained LLMs and ViTs (Sun et al., 2024; Darcet et al., 2024), where certain tokens from low semantic meaning regions, such as image background or the language punctuation, are allocated with high attention weights. Early work in LLMs identified the beginning-of-sequence (BOS) token as a consistent sink (Cancedda, 2024; Gu et al., 2024), later extended to show that sink tokens can emerge throughout the input (Yu et al., 2024). While these tokens can help with long-context modeling and quantization (Zhang et al., 2024; Ge et al., 2023; Xiao et al., 2023a), not all sink are beneficial, some reduce performance (Yu et al., 2024).

Visual attention also exhibits sinks (Sun et al., 2024) in models such as CLIP, and DINOv2 (Oquab et al., 2023). Visual sinks, however, are broadly considered not very useful, with majority of works focusing on their mitigation. Kang et al. (2025) identified visual attention sinks and redistribute their attention to semantically meaningful regions. Huang et al. (2024); Woo et al. (2024) propose to suppress the sink token to improve hallucination. Darcet et al. (2024) add registers in the beginning of the token sequence to absorb the information stored in sinks in ViT and result in clearer map for object discovery. In contrast, to the best of our knowledge, we are the first to empirically study and demonstrate that in LVLMs, visual attention sinks, propagated from ViT to LLM, play a crucial role.

## 7    CONCLUSION

Contrary to prior works, we show that *ViT sinks* in VLMs encode vital high-level visual context. Based on this insight, we introduce two effective methods to leverage them: a training-free repositioning strategy and a training-based framework, *DIYSink*, both validated across diverse benchmarks.

### REPRODUCIBILITY STATEMENT

To ensure the reproducibility of our work, we provide details of our analysis, methodology, and experiments. All experiments are conducted on publicly available benchmarks, as detailed in Sec. 5 and Sec. B.5. Our proposed framework, model configurations, and training details, including hyperparameters, are described in Sec. B.4. Furthermore, the specifics of our analysis are documented in the appendix, such as the definitions and thresholds used for identifying ViT sink tokens in Sec. A and Sec. B.1, and the prompts used for our GPT task clustering in Sec. B.2. Last but not least, as stated in Sec. 1, we will release all code and analysis scripts upon acceptance.

### ACKNOWLEDGEMENTS

This work was funded, in part, by the Vector Institute for AI, Canada CIFAR AI Chairs, NSERC Canada Research Chair (CRC), AML-TN UBC, and NSERC Discovery and Discovery Accelerator Supplement Grants. Resources used in preparing this research were provided, in part, by the Province of Ontario, the Government of Canada through CIFAR, the Digital Research Alliance of Canada[2], companies[3] sponsoring the Vector Institute, and Advanced Research Computing at the University of British Columbia. Additional hardware support was provided by John R. Evans Leaders Fund CFI grant and Compute Canada under the Resource Allocation Competition award.

---

[2]`alliance.can.ca`
[3]`https://vectorinstitute.ai/#partners`

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

## ABSTRACT

This document provides supplementary materials for our main submission. Section A presents additional analysis results on various LVLMs, including TinyLLaVA-0.5B & 3B, InternVL-2.5-4B, and LLaVA-7B. Also, we provide additional studies about the impact of ViT sinks in LLM, semantics of ViT sinks, and the emergence of ViT sink tokens during the training in this section. Section B describes the implementation details of our analysis, model designs, and benchmarks. Section C offers further quantitative analysis, including an ablation study of the Dual-MLP design, as well as inference weights from the reweighting module and the Chain-of-Thought (CoT) approach. Furthermore, we discuss the training and inference computation cost of our proposed frameworks. Section D provides qualitative results, including reasoning processes, model outputs, and word mapping results. Section E discusses the model's limitations and potential social impact.

## APPENDIX CONTENTS

## A  VIT SINKS IN LVLMS

### A.1  PROPAGATION OF VIT SINK TOKENS INTO LLM AND HIDDEN DIMENSION BREAKDOWN

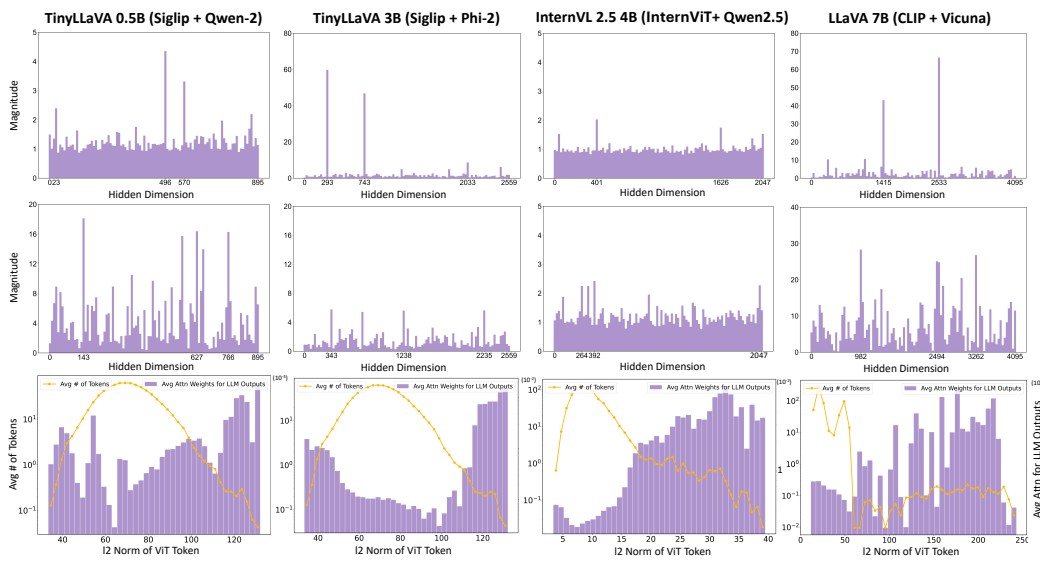

Figure A1: **Extra analysis results of the sink property.** We provide a dimension breakdown of LLM-emerged sinks (first row) and propagated ViT sinks (second row). Additionally, we examine the relationship between ViT token norms, calculated within the ViT, and the attention weights they receive from the LLM during output generation (last row). Results are averaged across 300 image-question pairs.

In Sec. 3, we analyze the properties of ViT sink tokens in large vision-language models (LVLMs), focusing initially on results from LLaVA-7B. To provide a comprehensive comparison, we extend our analysis to a diverse set of LVLMs incorporating various ViT backbones (e.g., CLIP, SigLIP, InternViT) and language models (e.g., Qwen-2, Qwen-2.5, Phi-2, Vicuna). Specifically, we examine TinyLLaVA-0.5B, TinyLLaVA-3B, InternVL-2.5-4B, and LLaVA-7B, with results presented in Fig. A1.

In Sec. 3, we observe a positive correlation between the norm of ViT tokens and the attention weights assigned by the LLM during output generation. We further investigate the generalizability of this observation. As shown in the last row of Fig. A1, across all evaluated models, we find that as the norm of ViT tokens increases, the corresponding attention from the LLM also increases during generation. This consistent trend reinforces our earlier observation of a positive correlation between ViT token norm and LLM attention. Additionally, by comparing the dimensional breakdown of LLM-emerged sinks (first row) and propagated ViT sinks (second row), we observe that the sets of highly activated hidden dimensions differ between the two sink types, which again support the observation mentioned

in Sec. 3. Note that the InternVL model exhibits notably weaker high-activation signals compared to other models. We attribute this to its use of visual features extracted from the final layer of the ViT, whereas the other models rely on features from the second last layer. Our findings suggest that the final ViT layer exhibits weaker sink activation signals relative to the second last layer. Note that all results are averaged across 300 image-question pairs.

## A.2 IMPACT OF VIT SINK TOKENS IN LLM

We further investigate the importance of ViT-propagated sinks by analyzing the average attention weights received by ViT sink tokens, as the results shown in Tab. A1. Specifically, we compute the average attention from output tokens to target tokens across 1000 image-question pairs. Our analysis reveals that, on average, ViT sink tokens receive attention weights that are comparable to or higher than those received by LLM-emerged sink tokens. Moreover, the attention weights assigned to ViT sink tokens are significantly higher, ranging from 3 to 10 times greater, than those of non-sink tokens, highlighting the important role of ViT sinks in guiding LLM output.

|  | TinyLLaVA 0.5B | TinyLLaVA 3B | InternVL 2.5 4B | LLaVA 7B |
|---|---|---|---|---|
| Propagated ViT Sink Attn | 0.0072 | 0.0329 | 0.0117 | 0.0113 |
| LLM-emerged Sink Attn | 0.0181 | 0.0367 | 0.0054 | 0.0127 |
| Non-sink Attn | 0.0013 | 0.0011 | 0.0032 | 0.0015 |

Table A1: Comparison of sink-related metrics across different models.

## A.3 SEMANTICS OF VIT SINK TOKENS

In Sec. 3.2, we use relevance map and word distribution to uncover the information captured in the VIT Sink. To further investigate semantics of ViT sink token, we conduct linear-probing analyses at the object level (object classification) and scene level (lighting, composition, and counting).

Table A2: Tasks definition, objectives, and data labeling strategy on the COCO dataset.

| Task | Objective | Data & Labeling (on COCO dataset) |
|---|---|---|
| **Obj class** | Test object class-sensitivity | Select images with only one instance from top 10 most frequent categories and label them with the object class. |
| **Lighting** | Measure awareness of overall scene brightness | Randomly select 1000 images; each augmented at 4 brightness scales (labels 0–3). |
| **Composition** | Determine whether sink tokens encode spatial quadrant | Select images with exactly one object from the top 5 most frequent categories; label = quadrant (TL/TR/BL/BR). |
| **Counting** | Evaluate scene-level detail via object counting | Select images with 1–4 instances from the top 5 most frequent categories; label = object count (1–4). |

**Analysis setup** For each task, we pair COCO images with ground-truth labels (e.g., object category, lighting-level). Given an input image, we extract its sink feature by averaging all sink-token embeddings from the vision encoder. For comparison, we randomly sample 5 non-sink tokens and average them to form a non-sink feature. We then train a simple linear classifier on these features: high validation accuracy indicates that the feature encodes the target information. We test four different tasks, object classification, lighting, composition, and counting, using the task definitions described in Tab. A2. Note that the number of data for each task: 4000 (training) and 1000 (testing).

**Discussion** As the results shown in Tab. A3, sink tokens from both CLIP and SigLIP outperform non-sink tokens by $20 - 30\%$ points on object classification and lighting tasks. They also lead in composition and counting, though the gains are smaller, consistent with the findings in Sec. 3.2, which show that sink tokens capture mostly global cues (e.g., object saliency, illumination) with minor local details (e.g., object position, count).

Table A3: Comparison of CLIP and SigLIP across various tasks.

| | Obj class | | Lighting | | Composition | | Counting | |
|---|---|---|---|---|---|---|---|---|
| | CLIP | SigLIP | CLIP | SigLIP | CLIP | SigLIP | CLIP | SigLIP |
| Random | 0.100 | 0.100 | 0.250 | 0.250 | 0.250 | 0.250 | 0.250 | 0.250 |
| Non-sink | 0.512 | 0.647 | 0.361 | 0.515 | 0.295 | 0.280 | 0.447 | 0.495 |
| Sink | **0.865** | **0.850** | **0.653** | **0.722** | **0.310** | **0.345** | **0.555** | **0.599** |

### A.4 THE EVOLUTION OF PROPAGATED VIT SINK TOKENS DURING LVLM TRAINING.

Throughout our training process, the vision encoder (ViT) is kept frozen during both the pretraining and fine-tuning stages. Consequently, the original ViT sink tokens remain static. This raises a key question: how do the corresponding propagated sink tokens, which are processed by the projector and language model, evolve at different training checkpoints?

To investigate this, we analyzed the hidden states of the propagated sink tokens across various training iterations. For a given image, we tracked the four hidden dimensions with the highest activation values. The results for our DIYSink 3B model are presented in Tab. A4. The results show that, during the projector pretraining stage, across training iterations, no single dimension exhibits disproportionately high values. However, in the early stages of fine-tuning, specific hidden dimensions emergeand attain much higher values than the others (e.g., $70.14$ in dim 490 in the finetuning of 0.5B model). Thus, it appears that ViT sink tokens begin to propagate during the early iterations of fine-tuning, and the set of highly activated dimensions then remains largely consistent throughout the rest of training.

Table A4: The behavior of the propogated VIT sink tokens across training progress of DIYSink-Qwen2-3B.

| | Pretraining | | | | Fine-tuning | | | |
|---|---|---|---|---|---|---|---|---|
| Training Iteration | 200 | 800 | 1400 | 2000 | 400 | 1200 | 2400 | 3600 |
| **(Dim idx: value)** | (2311, 9.2) | (1773, 4.2) | (599, 5.2) | (1097, 7.7) | (743, 44.0) | (743, 34.3) | (743, 34.2) | (743, 36.2) |
| | (393, 8.6) | (169, 3.8) | (1662, 5.2) | (1195, 5.6) | (293, 18.2) | (293, 17.3) | (293, 14.1) | (293, 25.3) |
| | (1254, 8.4) | (1799, 3.7) | (1787, 5.0) | (1625, 5.2) | (1948, 9.2) | (573, 13.3) | (618, 10.6) | (573, 10.5) |
| | (1796, 8.1) | (902, 3.6) | (2521, 4.9) | (599, 4.9) | (1546, 9.0) | (1546, 11.8) | (567, 8.0) | (1948, 9.0) |

## B IMPLEMENTATION DETAILS

### B.1 ANALYSIS DETAILS: VIT SINKS IN LLM

In Sec. 3.1, we mentioned that we use Eq. (1) to identify sink tokens. As described in the preliminaries, we use feature norms to identify ViT sinks and sink dimension values to identify all sinks within the LLM. The $\tau$ values used in the equation for each type of sink are provided in Tab. A6.

**Justification of $\tau$** For CLIP-based models, we observed a drop of token norms from a 3+ digit number (e.g., $177.7$) to a two-digit norm value (e.g., $47.5$). To further verify that the gap between sink and non-sink tokens is large, we train our DIYSink with CLIP+Qwen2-0.5B and SigLIP+Qwen2-0.5B using different thresholds, including 80, 90, 100, 110, 120. Across all thresholds, the performance variation on MME for both model are small (below $0.2\%$ change). This indicate that the model is unsensitive to the change of threshold, the gap is indeed large and we can probably split sink/non-sink tokens by simply setting a threshold. Same approach applied when finding $\tau$ for SigLIP and other vision backbones.

**Additional discussion of the adaptive, data-driven method for sink token selection** To further address the potential concern of manual tuning and enhance generalization across various architectures, we have studied an adaptive, data-driven method for sink token selection. Based on the characteristics of sink tokens, we know that if the visual tokens are sorted by their norm, a significant relative drop in norm will be observed between the last sink token and the first non-sink token. Thus, we can identify sink tokens by detecting the last significant relative drop (e.g., greater than $10\%$)

among the top-ranked tokens, allowing the model to adaptively determine the number of sink tokens for each model and dataset. We evaluated this adaptive method across four Large Vision-Language Model (LVLM) architectures, including training-from-scratch models like TinyLlava-0.5B (Qwen) and TinyLlava-3B (Phi), and training-free models like InternVL2.5-4B and Phi3.5V. The results on the MME benchmark is provided in Tab. A5. This table shows that the adaptive method achieves performance comparable to the fixed-threshold baseline, thereby proving its effectiveness by automating token selection without relying on a hard threshold.

Table A5: Comparison of the adaptive method for sink token selection on MME.

| Methods | ALL | Perception | Cognition | ComRea | NumCal | TextTrans | CodeRea |
|---|---|---|---|---|---|---|---|
| *Training-from-scratch* | | | | | | | |
| **Tinyllava 0.5B (Qwen2)** | 1381.10 | 1173.96 | 207.14 | 82.14 | 37.50 | 50.00 | 37.50 |
| DIYSink (CoT) hard | 1456.78 | 1179.28 | 277.50 | 85.00 | 60.00 | 80.00 | 52.50 |
| DIYSink (CoT) adaptive | 1455.77 | 1183.27 | 272.50 | 85.00 | 55.00 | 80.00 | 52.50 |
| **Tinyllava 3B (Phi)** | 1455.22 | 1193.43 | 261.79 | 99.29 | 55.00 | 50.00 | 57.50 |
| DIYSink (CoT) hard | 1523.18 | 1242.47 | 280.71 | 110.71 | 52.50 | 67.50 | 50.00 |
| DIYSink (CoT) adaptive | 1526.01 | 1236.01 | 290.00 | 115.00 | 55.00 | 67.50 | 52.50 |
| *Training-free* | | | | | | | |
| **InternVL2.5-4B** | 2332.16 | 1686.81 | 645.35 | 142.86 | 124.99 | 200.00 | 177.50 |
| + Sink-to-the-front (hard) | 2351.33 | 1706.33 | 645.00 | 145.00 | 125.00 | 200.00 | 175.00 |
| + Sink-to-the-front (adaptive) | 2357.18 | 1712.18 | 645.00 | 147.23 | 125.00 | 200.00 | 172.50 |
| **Phi3.5V** | 1887.94 | 1475.76 | 412.14 | 137.14 | 90.00 | 87.50 | 97.50 |
| + Sink-to-the-front (hard) | 1891.27 | 1457.35 | 433.92 | 136.42 | 97.50 | 87.50 | 112.50 |
| + Sink-to-the-front (adaptive) | 1889.28 | 1453.17 | 436.11 | 136.42 | 95.00 | 87.50 | 112.50 |

## B.2 ANALYSIS DETAILS: CLUSTERING

In Sec. 3.3, we introduce a task taxonomy based on query and image properties and investigate the information encoded in propagated ViT sinks and how the LLM interprets it. Specifically, we leverage GPT-4o to classify image-question pairs according to (1) image complexity, which measures the visual density and richness of the scene, and (2) query globalness, which assesses whether the question requires high-level contextual reasoning or fine-grained spatial understanding. We provide in Fig. A3 the detailed prompts used for GPT-based clustering to support reproducibility. We also conduct the human study for verifying the robustness of leveraging GPT as the classifier in the task in Sec. C.11.

## B.3 SINK TOKEN POSITION EMBEDDING DETAILS

To preserve the spatial layout of the image in both the training-free sink-to-the-front and train-from-scratch settings, we apply slightly different adjustments to the position embeddings of tokens. Below, we detail how sink and non-sink tokens are arranged and how their position IDs are assigned. We also include a figure to help illustrate, please refer to Fig. A2.

**Training-free:** as noted in Sec. 4.1 the main paper, we preserve the spatial structure by moving both the sink tokens and their original positional embeddings together. This ensures that the spatial layout encoded by the position embeddings remains intact, even though the tokens are reordered in the sequence.

**Train-from-scratch:** the model is retrained with sink tokens copied to the front, while their original positions are filled with constant vectors. This preserves the overall spatial grid and allows the model to learn to interpret the modified sequence without losing spatial consistency.

Importantly, sink tokens typically correspond to only 3–5 background tokens out of a much larger set of visual tokens (often 576 or more). Because they constitute such a small fraction of the sequence, repositioning them has minimal impact on the effective spatial layout of the image. This also helps explain why models remain stable despite the reordering.

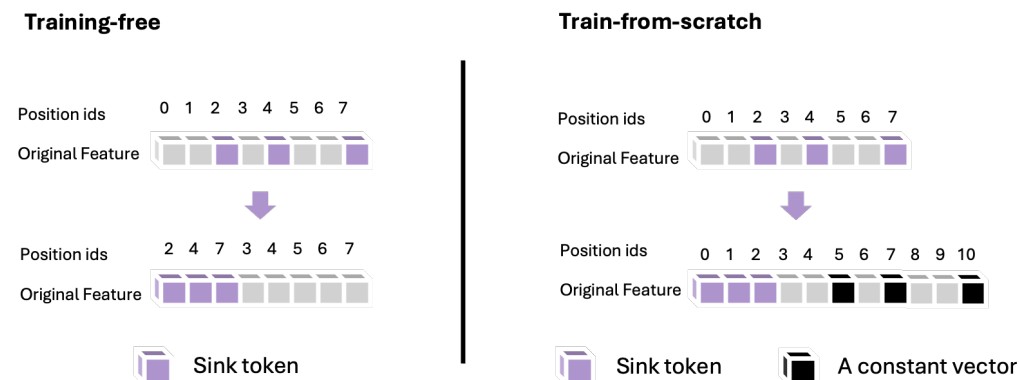

Figure A2: Sink tokens reordering and their respective position ids for training-free and train-from-scratch settings

|  | TinyLLaVA 0.5B | TinyLLaVA 3B | InternVL 2.5 4B | LLaVA 7B |
|---|---|---|---|---|
| $\tau$ (ViT Sink) | 120 | 120 | 35 | 100 |
| $\tau$ (Propagated ViT Sink) | 4 | 5 | 3 | 4 |
| $\tau$ (LLM-emerged Sink) | 5 | 25 | 4 | 20 |

Table A6: Implementation details of sink characteristic function.

## B.4 FRAMEWORK DETAILS

The Dual-MLP connector is first separately pretrained and subsequently fine-tuned jointly with the LLM, using the same hyperparameter settings and training examples as those employed in the LLaVA framework Liu et al. (2023a).

For the Reweighting MLP module, training samples are drawn from the training sets of PIXMO Deitke et al. (2024) and GeoQA Chen et al. (2021), which are distinct from our evaluation benchmarks. We use a stratified sampling strategy based on model scale: 120 examples for 0.5B and 3B models, 240 examples for the 7B model. In each case, the dataset is balanced across three task types: Local tasks (color recognition and counting), Global tasks (geographic reasoning), Mixed tasks (chart and table understanding), with each category comprising one-third of the total samples. We fine-tune only this component while keeping all other modules frozen, in order to prevent additional information leakage and ensure a fair evaluation. The Reweighting MLP is trained for 10 epochs on the respective data, and optimized using a learning rate of 1e-2 and a global batch size of 20. All the training are done using 4 Nvidia A40.

## B.5 BENCHMARKS DETAILS

We evaluate our method on a wide range of benchmarks including general benchmarks used in LLaVA (or LLaVA eval) Liu et al. (2023c), GQA Hudson & Manning (2019), ScienceQA Lu et al. (2022), TextVQA Singh et al. (2019a), MMMU Yue et al. (2024), fine-grained benchmark like MME Fu et al. (2023) and mathematical reasoning benchmark MathVista Lu et al. (2024b) to examine the influence of *DIYSink* on LVLMs following the common protocols of these benchmarks. The **GQA** dataset is a large-scale, balanced visual question answering benchmark designed to test and diagnose visual reasoning and scene understanding through compositional, semantically structured questions over real-world images. We evaluated on the test set which contains 12,578 examples. The **ScienceQA** dataset is a large-scale, multimodal benchmark comprising 21,208 multimodal multiple-choice science questions, with majority featuring images and text contexts. Each data point is annotated with detailed lectures and explanations to support chain-of-thought reasoning across diverse subjects like natural, social, and language sciences. We evaluated on the test set which contains 4,241 examples. The **TextVQA** dataset requires model to understand the text in the image and reason about it to answer questions. We evaluated on the testset which contains 5,000 examples. The **MMMU** (Massive

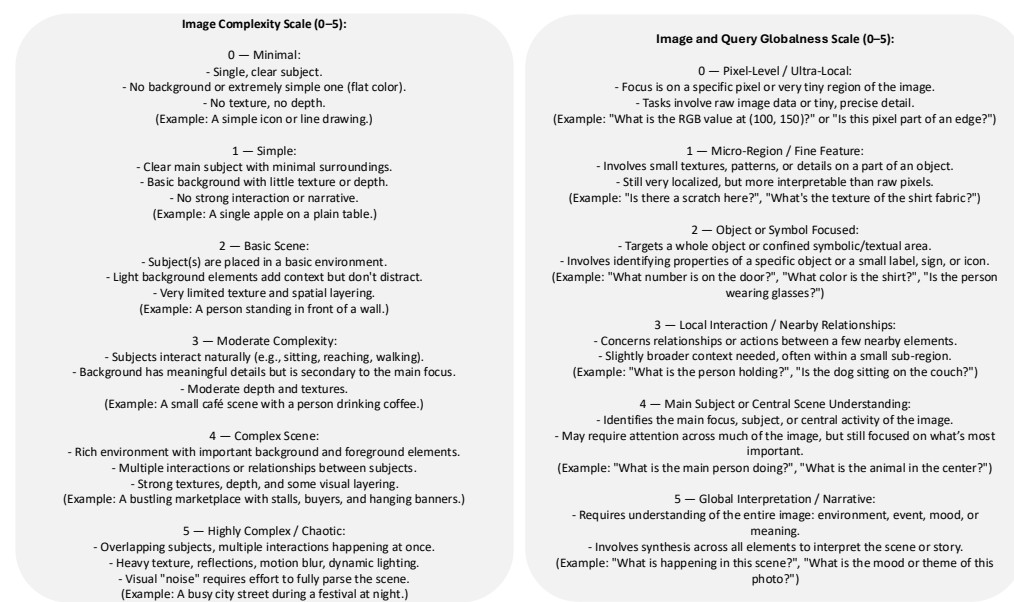

Figure A3: **Details GPT-4o prompt for clustering experiments.**

Table A7: **Additional comparison of Qwen2.5.** ReW and CoT denote our reweighting module and Chain-of-Thought prompting, respectively. Green indicates increase over the corresponding baseline. † denotes models are fine-tuned on original LLaVA fine-tuning datasets.

| Methods | LLaVA eval | | | | | MME | | | | | MathVista | | | |
|---|---|---|---|---|---|---|---|---|---|---|---|---|---|---|
| | **AVG** | GQA | SQA | TextVQA | MMMU | **All** | Cog. | ComRea | TextTrans | **CodeRea** | **All** | ALG | GEO | LOG |
| **TinyLLaVA-0.5B-SigLIP-Qwen2.5-0.5B** | | | | | | | | | | | | | | |
| Baseline† | 49.57 | **58.62** | 59.89 | 47.78 | 32.00 | 1506.68 | 215.00 | 80.00 | 47.50 | 47.50 | 24.40 | 24.91 | 23.85 | 5.41 |
| *DIYSink* (self-CoT) | 50.02 $_{+0.45}$ | 58.54 | **60.09** | 48.64 | 32.80 | 1522.45 $_{+15.77}$ | **241.07** | 78.57 | **50.00** | **55.00** | 26.00 $_{+1.60}$ | 25.62 | 24.27 | **16.22** |
| *DIYSink* (ReW) | 50.34 $_{+0.77}$ | 58.45 | 60.04 | **48.88** | **34.00** | 1526.69 $_{+20.01}$ | 240.71 | **80.71** | **50.00** | 52.50 | **26.90** $_{+2.50}$ | **29.54** | **28.03** | **16.22** |
| **TinyLLaVA-3.1B-SigLIP-Qwen2.5-3B** | | | | | | | | | | | | | | |
| Baseline† | 70.22 | 62.47 | 74.12 | 58.35 | **40.20** | 1740.01 | 293.21 | 115.71 | 65.00 | 52.50 | 30.40 | 37.01 | 38.08 | 18.92 |
| *DIYSink* (CoT) | 70.98 $_{+0.76}$ | **63.62** | 75.41 | 58.54 | 39.60 | 1758.47 $_{+18.46}$ | **302.14** | 127.14 | 70.00 | 47.50 | 33.40 $_{+3.00}$ | **41.64** | 42.68 | **24.32** |
| *DIYSink* (ReW) | 71.65 $_{+1.43}$ | 63.58 | **76.20** | **60.72** | 39.00 | 1761.41 $_{+21.40}$ | 299.28 | **129.28** | 70.00 | 47.50 | 33.10 $_{+2.70}$ | 41.63 | **43.10** | 24.32 |

Multi-discipline Multimodal Understanding and Reasoning) benchmark is a comprehensive dataset comprising 11.5K college-level multimodal questions across six disciplines and 30 subjects, designed to evaluate models' expert-level visual perceptual abilities and deliberate reasoning with subject-specific knowledge. We evaluated on the testset which contains 900 examples. The **MME** dataset measures the perception and cognition abilities of LVLMs. Specifically, it assesses the models' capability in recognizing the specific object, such as its existence, count, position and color, and then compositing the perception information and the knowledge in LLM to deduce more complex answers. **MathVista** is a comprehensive benchmark designed to evaluate the mathematical reasoning abilities of foundation models within visual contexts. It comprises 6,141 examples drawn from 28 existing multimodal datasets and three newly created ones–IQTest, FunctionQA, and PaperQA–requiring deep visual understanding and compositional reasoning skills. We evaluated on testmini which contains 1000 examples.

## C    EXTRA QUANTITATIVE RESULTS

### C.1    SINK POSITION ABLATION

The primary motivation for repositioning sink tokens is to amplify their impact during output generation, which we can achieve by leveraging two positional biases in LLMs:

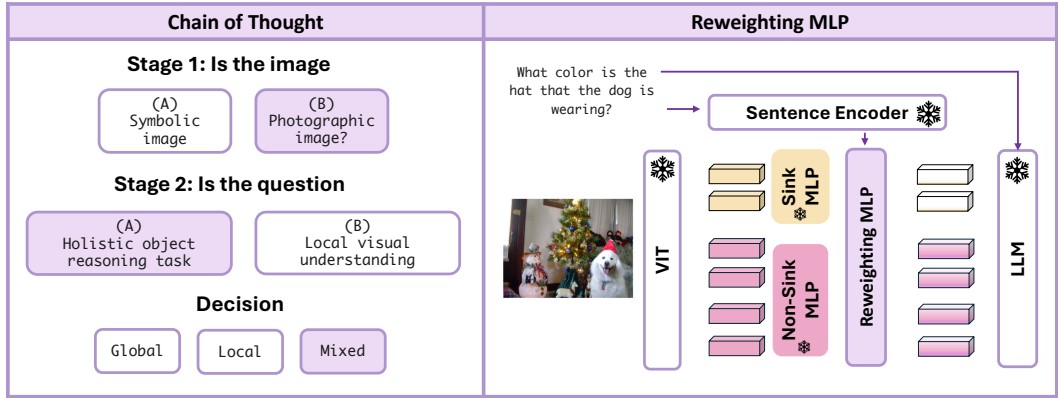

Figure A4: Illustration of our Chain of Thought inference and Reweighting MLP module.

**Causal Attention Bias**: In autoregressive Transformers, each token attends only to its predecessors, amplifying the influence of early tokens because information flows forward but not backward. By moving sink tokens, together with their original positional embeddings, to the front of the sequence, we exploit this bias.

**RoPE Recency Bias**: Rotary Position Embeddings introduce a mild "recency" effect Wang et al. (2024): attention weights decay as the relative distance between query and key increases, causing the model to favor tokens nearer the end of the sequence . If we reposition tokens and reassign their positional embeddings, we can leverage this effect.

Although both biases shape attention, our empirical study, in Tab. A8 and Tab. A9, shows that the causal attention bias dominates the RoPE recency bias, enabling front-positioned tokens to exert greater influence. Based on these findings, placing sink tokens at the front ensures they receive stronger, earlier attention, resulting in consistently better performance than appending them at the end.

Table A8: Sink token position ablation with training-free models.

| Methods | GQA | SQA-image | TextVQA | MMMU | MME | MathVista |
|---|---|---|---|---|---|---|
| **InternVL2.5-4B** | 62.1 | 96.2 | **73.5** | 50.2 | 2332.2 | 62.9 |
| ≫ sink-to-the-end | 62.2 | **96.2** | 73.3 | 50.3 | 2331.5 | 60.2 |
| ≫ sink-to-the-front | **62.4** | 96.1 | 73.2 | **50.4** | **2351.3** | **63.3** |
| **Phi 3.5 V** | 63.5 | 91.3 | **65.2** | 43.2 | 1887.9 | 43.1 |
| ≫ sink-to-the-end | 63.5 | 91.0 | 64.2 | 43.8 | 1875.9 | 43.3 |
| ≫ sink-to-the-front | **63.5** | **91.3** | 64.1 | **44.0** | **1891.3** | **43.5** |

Table A9: Sink token position ablation with train-from-scratch models.

| Methods | GQA | SQA-image | TextVQA | MMMU | MME | MathVista |
|---|---|---|---|---|---|---|
| **Tinyllava 0.5B (Qwen)** | 58.0 | 57.8 | 47.3 | 30.3 | 1381.1 | 24.3 |
| » Dual MLP with sink-to-the-end | 56.9 | 60.6 | 47.6 | 30.4 | 1433.9 | 24.6 |
| » Dual MLP with sink-to-the-front | **58.3** | **60.6** | **48.4** | **30.8** | **1439.2** | **24.8** |
| **Tinyllava 3B (Phi)** | 51.1 | 68.5 | 41.2 | 33.5 | 1455.2 | 25.9 |
| » Dual MLP with sink-to-the-end | 49.8 | 65.9 | 37.6 | 34.3 | 1318.0 | 26.8 |
| » Dual MLP with sink-to-the-front | **60.0** | **70.8** | **47.7** | **34.8** | **1679.2** | **28.4** |

## C.2 DUAL-MLP ABLATION

From the results in Fig. 5 B, we observe a performance gap between the upper-bound achieved using designated tokens for global and local tasks and the baseline model's actual performance. This suggests that the LLM struggles to leverage sink and non-sink tokens simultaneously without interference. To address this, we first propose using Dual-MLP Projection Layers to independently process sink and non-sink ViT tokens, allowing for appropriate projections that account for their distinct distributions.

Table A10: Performance of the Dual-MLP model (default as sink-to-the-front) and the sink-only and non-sink only variant, where we only input the respective type of tokens into LLM during inference.

| Methods | LLaVA_eval | | MME | | | MathVista | | |
|---|---|---|---|---|---|---|---|---|
| | MMMU-val | SQA-img | Position | OCR | CodeRea | GEO | LOG | ALG |
| LLaVA 7B | 35.80 | 68.91 | 133.33 | 125.00 | 42.50 | 20.92 | 13.51 | 25.27 |
| Dual-MLP | | | | | | | | |
| Ours (default sink-to-the-front) | **36.90** | **69.61** | 136.67 | 130.00 | 55.00 | 23.43 | 13.51 | 26.69 |
| Ours (sink) | 33.40 | 65.29 | 63.33 | 130.00 | **72.50** | **28.87** | **21.62** | **30.96** |
| Ours (non-sink) | 35.90 | 68.86 | **141.67** | **132.50** | 57.50 | 21.76 | 13.51 | 23.84 |

Table A11: Comparison of LVLMs with different types of connectors. DIYSink improves TinyLLaVA 0.5B (TL) across both Q-Former and Resampler connector variants.

| Method | LLaVA eval | | | | MME | | | | | MathVista | | | |
|---|---|---|---|---|---|---|---|---|---|---|---|---|---|
| | GQA | SQA-image | TextVQA | MMMU-val | ALL | Cognition | ComRea | TextTrans | CodeRea | ALL | ALG | GEO | LOG |
| TL - Q-former | 47.75 | 55.78 | 33.53 | **31.00** | 1160.33 | 221.43 | 71.43 | 47.50 | **50.00** | 23.30 | 24.56 | 23.01 | **13.51** |
| TL - Q-former (COT) | **48.01** | **56.62** | 33.93 | 30.00 | **1173.84** | **257.86** | **77.86** | **75.00** | 47.50 | **23.80** | **25.98** | **25.10** | 13.51 |
| TL - Resampler | 53.65 | **61.18** | **45.30** | 32.00 | 1403.31 | 234.29 | 84.29 | 50.00 | 50.00 | 23.60 | 21.00 | 20.08 | **24.32** |
| TL - Resampler (COT) | **53.75** | 60.93 | 45.12 | **32.00** | **1427.11** | **237.50** | **85.00** | **50.00** | 50.00 | **24.00** | **22.42** | **22.18** | 21.62 |

With the connector fixed, we present an additional experiment to further examine whether the performance gap among the three variants, sink-to-the-front, sink-only, and non-sink persist. Similar to Sec. 3.3, we modify our Dual-MLP model to create the three variants of model by inputting only the respective tokens into the LLM during inference. As shown in Sec. C.2, the default dual-mlp sink-to-the-front consistently outperform the baseline across different tasks, showing the effectiveness of Dual-MLP module. In the mean time, we observe sink and non-sink only variant excels on different subsets of tasks, which indicate that LLM still struggles to jointly leverage sink and non-sink without interfacing each other. Following this, we design the Dynamic Token Selection Modules, as illustrated in Sec. 4.2 and present in Fig. A4 to dynamic reweight the tokens input into LLM in order to better approximate the performance upper bound of the sink and non-sink variants.

## C.3 CONNECTOR TYPE ABLATION

To further investigate the impact of connector architectures, we studied two cross-attention-based connectors: the Q-former and the Perceiver Resampler Jaegle et al. (2021). The results are provided in Tab. A11. We implemented these connectors within the TinyLLaVA-Qwen2-0.5B model architecture, replacing the standard MLP connector, and trained the models from scratch to establish baselines. In both architectures, the connector takes 728 visual tokens as input and outputs a compressed representation of 128 tokens via cross-attention. We then applied our DIYSink method. For a fair comparison with the baseline, we maintained the total output token count at 128 by setting the sink token connector (query/latent) size to 2 and the non-sink token size to 126. Following dual-connector training and chain-of-thought inference, the results demonstrate that both model variants with Q-former and the Perceiver Resampler achieve consistent improvements on the MME and MathVista benchmarks. Overall scores across general benchmarks also increased for both models. This indicates that use ViT sink tokens correctly is crucial for enhancing the reasoning capabilities of Large Vision-Language Models (LVLMs) with either cross-attention-based connectors or MLP connector.

## C.4 REWEIGHTING MLP INFERENCE WEIGHTS

We provided the inference weight for each benchmark we evaluated in the model to further validate our finding that *sink and non-sink tokens are beneficial to different tasks and reweighting them can help model inference.*

As we can see from Fig. A5, the model learned to assign more weights to global tasks for tasks like geometry reasoning, algebraic reasoning, arithmetic reasoning, code reasoning and commonsense reasoning, which require abstract context-level information and reasoning. For tasks like MME-

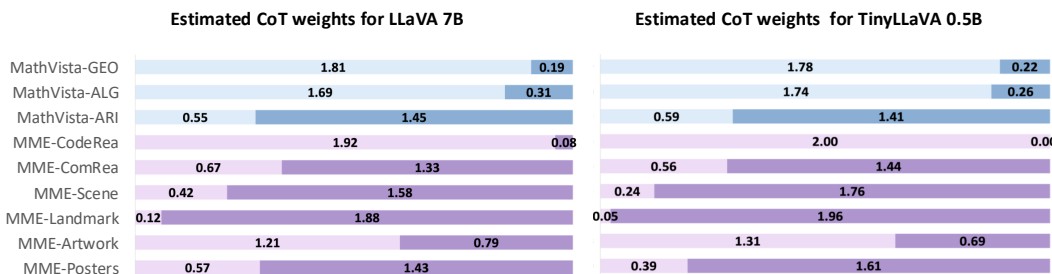

Figure A5: Output weights from the reweighting mlp layer. For each plot, the number on the left are the weight assigned to sink tokens and the number on the right are the weights assigned to non-sink tokens

Figure A6: Estimated CoT weights. For each plot, the number on the left are the weight assigned to sink tokens and the number on the right are the weights assigned to non-sink tokens.

Artwork and MME-Posters, which require more local information to detect the artwork or understand the poster, the model learned to assign more weight to non-sink tokens that contains detailed local information. Additionally, we analyze the selection behavior of the CoT approach. To enable quantitative analysis, we define global weights and local weights to represent the average CoT selection results. Specifically, in the sink-only case, the global weight is set to 2 and the local weight to 0; in the non-sink-only case, the global weight is 0 and the local weight is 2; and when both are used, each receives a weight of 1. This formulation allows us to compute the average global and local weights for each task. The results are presented in Fig. A6. Notably, the overall trends closely align with the reweighting MLP results. For reasoning-intensive tasks such as logical and geometric reasoning, the model predominantly leverages ViT sink tokens.

## C.5 PERFORMANCE ANALYSIS ON DENSE AND COMPOSITIONAL SCENES

As described in Sec. 3.2, we evaluate the impact of ViT sink tokens in LLM across global, local, and mixed tasks. Dense compositional scenes naturally fall under the local and mixed categories. As shown in Fig. 5(b), using only sink tokens leads to a performance drop on these tasks, which supports our key finding: "ViT sink tokens excel at global tasks, but fine-grained, localized tasks may require non-sink tokens, alone or with sink tokens, to capture detail.". To address this limitation, we introduce DIYSink, a dynamic token-selection framework that chooses between sink tokens, non-sink tokens, or both, depending on the specific task, thereby improving performance across varied scenarios. To further examine this finding, we evaluated three compositional reasoning benchmarks: (1) Spatial Relation Understanding (BLINK): Probes a model's ability to infer relative object locations. (2, 3) Position and Counting Tasks (MME): Requires the model to locate or enumerate objects in multi-object scenes. The comparison results are shown Tab. A12. We find that the baseline with only sink tokens underperforms compared to baseline which uses all tokens, reaffirming our observation. On the other hand, our DIYsink yields consistent gains across all architectures on compositional tasks by dynamically selecting between sink/non-sink or both token types based on the visual content and the query, demonstrating the effectiveness of the proposed approach.

Table A12: Model Performance Comparison Across Different Tasks. Tiny (0.5B) refers to TinyLLaVA-0.5B-SigLIP-Qwen2-0.5B and LLaVA (7B) refers to LLaVA-7B.

| | BLINK SpaRel | | MME position | | MME count | |
|---|---|---|---|---|---|---|
| Method | Tiny (0.5B) | LLaVA (7B) | Tiny (0.5B) | LLaVA (7B) | Tiny (0.5B) | LLaVA (7B) |
| Baseline | 53.85 | 67.13 | 101.66 | 133.33 | 101.66 | 160.00 |
| Baseline (Sink Only) | 53.85 | 51.05 | 63.33 | 55.00 | 98.33 | 60.00 |
| DIYSink (CoT) | 53.15 | 64.34 | 123.33 | **141.67** | 110.00 | **165.00** |
| DIYSink (ReW) | **53.85** | **68.53** | 123.33 | 131.66 | **113.33** | 160.00 |

## C.6 ANALYSIS OF THE TRAINING AND INFERENCE COMPUTATIONAL COST

We profiled both training and inference cost of three explored DIYSink variants, Dual-projector only, Dual+CoT routing, and Dual+ReW routing. Note that we use SigLIP + Qwen2-0.5B as the base model and measure the cost on Nvidia L40 (48GB). The results are shown in Tab. A13. All measurements use a batch size of 1 on a single GPU for memory and inference phase profiling, and 4 GPUs for training-time profiling. Peak memory costs are reported for GPU and CPU cost measurement.

**Parameter overhead** ($+0.2\%$). This stems from adding an additional MLP projector for sink vs. non-sink token separation, and, in the ReW variant, an extra MLP to compute per-token reweighting coefficients.

**Training overhead.** Since we pretrain the projector for sink and nonsink tokens separately, the total training time cost increases (so as FLOPs) roughly by one hour on 4xL40s. We note that this can be reduced greatly by parallelly pretraining two MLP projectors.

**Inference latency.** Our two final models, Dual+CoT and Dual+ReW, exhibit distinct trade-offs. Dual+ReW incurs an additional GPU overhead and increases per-instance latency by roughly 0.01s. In contrast, Dual+CoT leverages the model's own multi-turn conversation routing, adding about 1.6s per instance but without any extra parameters. Both methods deliver solid performance gains over the baseline (see Tab. 2) and can be chosen according to users' latency and resource requirements. As for the tight-latency scenarios, we can adopt the Dual+ReW routing which captures the majority of DIYSink's benefits while adding only 0.01s of extra inference time.

Table A13: Resource Usage for Tinyllava SigLIP+Qwen2-0.5B.

| Training phase - Tinyllava SigLIP+Qwen2-0.5B | | | | | |
|---|---|---|---|---|---|
| | Parameters | Time cost | FLOPs | GPU mem (peak) 1bs | CPU mem (peak) |
| Baseline | 924.09 M | 41981.97 | 4.3784E+15 | 14.75 GB | 32.52 GB |
| w/ Dual | 925.93 M | 46120.79 | 4.8417E+15 | 17.07 GB | 34.93 GB |
| w/ Dual w/ CoT | 925.93 M | 46120.79 | 4.8417E+15 | 17.07 GB | 34.93 GB |
| w/ Dual w/ ReW | 925.98 M | 46437.70 | 4.8421E+15 | 17.07 GB | 34.93 GB |

| Testing phase - (test on GQA) | | | | | |
|---|---|---|---|---|---|
| | Parameters | Time cost (per instance) | FLOPs | GPU mem (peak) BS=1 | CPU mem (peak) |
| Baseline | 924.09 M | 0.0928 | 1.8617E+09 | 7.33 GB | 22.67 GB |
| w/ Dual | 925.93 M | 0.0947 | 1.8620E+09 | 7.89 GB | 22.78 GB |
| w/ Dual w/ CoT | 925.93 M | 0.2518 | 6.8232E+09 | 7.89 GB | 22.78 GB |
| w/ Dual w/ ReW | 925.98 M | 0.1057 | 1.9442E+09 | 8.42 GB | 23.05 GB |

## C.7 ANALYSIS OF THE SINK/NON-SINK TOKEN CLASSIFICATION ACCURACY FOR DIYSINK

As shown in the Table A14, sink and non-sink tokens behave very differently on global versus local tasks. When all local questions are mistakenly treated as global, as in the Ours (sink) setting, where only sink tokens are used, the performance on local tasks drops sharply, often falling well below the baseline. Conversely, in the Ours (nonsink) setting, using only non-sink tokens causes substantial degradation on global tasks, when they are misclassified as local.

Table A14: Model performance on Local and Global tasks. Local represent tasks that non sink tokens work well and Global represent tasks that sink tokens work well. Sink and non-sink tokens behave very differently on global vs. local tasks. We bold the numbers where DIYSink surpass baseline as we are comparing with baseline. MV. stands for MathVista.

| Methods | Local | | | Global | | |
|---|---|---|---|---|---|---|
| | GQA | SQA-image | TextVQA | MME Cog. | MV. SciRea | MV. NUM |
| *Tinyllava 0.5B (Qwen)* | | | | | | |
| Baseline | 57.98 | 57.76 | 47.32 | 207.14 | 36.89 | 11.11 |
| Ours (sink) | 39.53 | 53.94 | 30.10 | 270.71 | 37.70 | 17.36 |
| Ours (nonsink) | 58.07 | 60.09 | 48.28 | 220.00 | 36.89 | 11.11 |
| DIYSink (CoT) | **58.14** | **61.08** | 46.14 | **277.50** | 37.70 | 11.81 |
| DIYSink (ReW) | 57.75 | 60.24 | **47.42** | 229.64 | **38.52** | **14.58** |
| *Tinyllava 3B (Phi)* | | | | | | |
| Baseline | 51.09 | 68.47 | 41.15 | 261.79 | 45.90 | 13.89 |
| Ours (sink) | 39.13 | 65.79 | 35.92 | 315.71 | 45.90 | 22.22 |
| Ours (nonsink) | 60.02 | 70.25 | 47.51 | 269.64 | 43.44 | 19.44 |
| DIYSink (CoT) | **59.99** | 68.57 | 46.22 | **280.71** | **47.54** | 13.89 |
| DIYSink (ReW) | 59.79 | **70.50** | **50.77** | 265.71 | 42.62 | **19.44** |

However, as shown in the DIYSink (CoT) and DITSink (ReW) row of Table A14, we observe consistent improvement in downstream tasks with our dynamic selection module. This demonstrates that DIYSink can adaptively choose the suitable token type for each task and approximate maximum performance, effectively leveraging the strengths of both sink and non-sink tokens. Note that when performing CoT or reweighting in DIYSink, GPT is not involved in the decision-making. GPT classification is only used for the analysis in Section 3 of the main paper.

## C.8 ANALYSIS OF USING CHAIN-OF-THOUGHT SELECTION MECHANISM AS THE SOFT SELECTION

In Sec. 4.2, we propose both a Chain-of-Thought (CoT) approach and a learnable reweighting (ReW) mechanism for sink token selection. While the CoT approach performs hard selection by choosing discrete inference modes (sink-only, non-sink-only, or both), the ReW mechanism already serves as a soft selection mechanism by assigning continuous weights to sink and non-sink tokens. Here, we additionally explore converting the CoT procedure itself into a soft selector. Specifically, for a given task, we first run the initial 10 VQA samples using the standard CoT to obtain the empirical distribution of sink and non-sink token usage. For instance, in a code reasoning task, if the first ten samples yield sink-only inference (6 samples), non-sink-only inference (2 samples), and both (2 samples), the resulting soft weighting values are calculated as $w_{sink} = (6 + 2)/10 = 0.8$ and $w_{non\text{-}sink} = (2 + 2)/10 = 0.4$. We then rerun the inference for that task using these soft reweighting values. As shown in Table A15, the performance of the Soft-CoT variant is on par with the original CoT. While the overall score of Soft-CoT is slightly higher than the original CoT in some settings, the performance on the cognition subtask can drop slightly, suggesting a trade-off between global and task-specific optimization.

## C.9 PARAMETER OVERHEAD OF THE DUAL-MLP MODULE

While the dual-MLP connector doubles the parameters of the original MLP, the connector constitutes only a very small portion of the overall LVLM. For example, in TinyLLaVA-0.5B, the dual-MLP module introduces only approximately 0.2% additional parameters (1.8M out of ∼1B total), making it unlikely that the observed improvements are driven solely by the parameter increase. To further disentangle this effect, we evaluate a baseline where the MLP depth is doubled (*w/ doublemlp*) while keeping a single connector pathway. As shown in Table A16, the doubled-MLP baseline performs comparably to, or only slightly above, the original single-MLP baseline across all three benchmarks, yet it consistently remains below both DIYSink variants (CoT and ReW). This confirms that the

Table A15: Comparison of DIYSink selection mechanisms on TinyLLaVA-0.5B (Qwen2) and TinyLLaVA-3B (Phi). **Bold** indicates the best result per model group. ReW denotes the learnable reweighting mechanism, CoT denotes the hard chain-of-thought selection, and Soft-CoT denotes the proposed soft chain-of-thought variant.

| Methods | ALL | Perception | Cognition | ComRea | TextTrans | CodeRea |
|---|---|---|---|---|---|---|
| *TinyLLaVA-0.5B (Qwen2)* | | | | | | |
| Baseline | 1381.10 | 1173.96 | 207.14 | 82.14 | 50.00 | 37.50 |
| DIYSink (ReW) | 1451.87 | 1222.23 | 229.64 | 87.14 | 50.00 | 37.50 |
| DIYSink (CoT) | **1456.78** | 1179.28 | **277.50** | 85.00 | **80.00** | **52.50** |
| DIYSink (Soft-CoT) | 1456.32 | **1238.46** | 217.86 | **87.86** | 50.00 | 32.50 |
| *TinyLLaVA-3B (Phi)* | | | | | | |
| Baseline | 1455.22 | 1193.43 | 261.79 | 99.29 | 50.00 | **57.50** |
| DIYSink (ReW) | **1682.41** | **1416.70** | 265.71 | 115.71 | 57.50 | 45.00 |
| DIYSink (CoT) | 1523.18 | 1242.47 | **280.71** | 110.71 | **67.50** | 50.00 |
| DIYSink (Soft-CoT) | 1579.85 | 1310.21 | 269.64 | **117.14** | 50.00 | 47.50 |

Table A16: Ablation on parameter overhead. "w/ doublemlp" doubles the MLP connector depth while maintaining a single pathway. Our dual-MLP module adds only ∼0.2% parameters yet yields consistent improvements, indicating gains arise from the design rather than increased capacity. **Bold** indicates the best result per column.

| Model | LLaVA Eval | MME | MathVista |
|---|---|---|---|
| TinyLLaVA 0.5B (Qwen2.5) | 49.57 | 1506.68 | 24.40 |
| TinyLLaVA 0.5B (Qwen2.5) w/ doublemlp | 50.18 | 1500.95 | 25.10 |
| Ours (CoT) | 50.02 | 1522.45 | 26.00 |
| Ours (ReW) | **50.34** | **1526.69** | **26.90** |

performance gains of DIYSink stem from its architectural design, separately routing sink and non-sink tokens, rather than from simply adding more parameters.

## C.10 Efficiency Analysis of ViT Sink Tokens

We further explore the efficiency implications of using only the small set of ViT sink tokens for inference. As shown in Table A17, we run baseline models by feeding only the ViT sink tokens into the LLM, which constitute approximately $0.5\%$ of the total visual tokens, and evaluate on reasoning-heavy subtasks from MME and MathVista. Despite using a drastically reduced set of visual tokens, sink-only inference yields consistent improvements on high-level global reasoning tasks. On MME, we observe gains in numerical calculation, text translation, and code reasoning; on MathVista, improvements are seen in algebraic, geometry, and scientific reasoning task. These results reinforce our finding that ViT sink tokens encode highly abstract, global visual information. Because these tokens capture distilled semantics most relevant for global reasoning, they enable efficient inference with improved performance on reasoning-centric benchmarks, highlighting their potential for token pruning and lightweight inference strategies.

## C.11 Human Study of Task Taxonomy using GPT

To validate the consistency of our GPT-based task taxonomy, we conducted a human verification study involving 12 evaluators who assessed 84 VQA pairs using the same criteria provided to GPT-4o. To first establish a reliability ceiling for this inherently subjective task, two independent annotators labeled the same set of 14 VQA pairs, yielding Cohen's Kappa scores of 0.689 for Image Complexity and 0.583 for Query Globalness (Table A18). These scores indicate that even human experts reach only "substantial" rather than "near-perfect" agreement, reflecting the intrinsic ambiguity of these dimensions.

Table A17: **Efficiency Analysis** Inference using only ViT sink tokens (∼0.5% of visual tokens). Sink-only inference consistently improves performance on high-level reasoning subtasks across MME and MathVista. **Bold** indicates the best result per model group.

| Methods | MME | | | MathVista | | | |
|---|---|---|---|---|---|---|---|
| | NumCal | TextTrans | CodeRea | ALG | GEO | SCI | GPS |
| TinyLLaVA-0.5B-Qwen2 | 37.50 | 50.00 | 37.50 | 22.42 | 20.92 | 36.89 | 22.12 |
| + sink only | **50.00** | **65.00** | **50.00** | **25.27** | **23.85** | **38.52** | **25.96** |
| LLaVA 7B | 55.00 | 55.00 | 42.50 | 25.27 | 20.92 | 39.34 | 21.15 |
| + sink only | **65.00** | **95.00** | **65.00** | **32.03** | **30.13** | **43.44** | **34.13** |

Table A18: Inter-annotator agreement for the task taxonomy. Cohen's Kappa and correlation coefficients are reported for human–human and human–GPT-4o comparisons across Image Complexity and Query Globalness dimensions.

| | Image Complexity | Query Globalness | Correlation Coefficient |
|---|---|---|---|
| Human vs Human | 0.6897 | 0.5826 | 0.6667 |
| Human vs GPT | 0.5263 | 0.4267 | 0.5120 |

Against this human baseline, GPT demonstrates robust alignment. When comparing continuous decimal scores with a 0.5 tolerance, machine–human agreement achieves Kappa scores of 0.526 (moderate) for Image Complexity and 0.427 (moderate) for Query Globalness, with a combined correlation coefficient of 0.512 compared to 0.667 for the human–human pair. These results confirm that GPT captures the underlying difficulty distribution with a reliability approaching that of human annotators, supporting its use as a scalable proxy for task classification.

## C.12 EVALUATION ON HALLUCINATION-BASED BENCHMARK

To further demonstrate the effectiveness of our methods, we extend our evaluation to include the POPE benchmark, which targets object hallucination by probing whether specific objects exist in an image. As shown in Table A19, our trained models (CoT and ReW) consistently improve over their respective baselines across all four model scales. We attribute this to our method's ability to dynamically reweight toward local non-sink tokens. Since POPE focuses on object existence, a localized task, this reweighting allows the model to better attend to specific object details, reducing hallucination.

## C.13 STATISTICAL ANALYSIS ON MODEL PERFORMANCE SIGNIFICANCY

To assess the statistical significance of both the training-free sink-to-the-front method and the train-from-scratch DIYSink approach, we conduct McNemar's test McNemar (1947); Raschka (1811) to compare each method against its corresponding baseline.

For the train-from-scratch setting, as shown in the right table of Table A20, we compute McNemar's test p-values over all benchmarks reported in Table 2, comparing DIYSink (ReW) with the baseline while assigning equal weight to each benchmark. The results are statistically significant for three out of four models. LLaVA-7B yields a p-value of 0.14, which still shows weak significance.

For the training-free methods, as discussed in Section 3, sink tokens benefit the model primarily on tasks requiring high-level, abstract reasoning (the Global and Mixed categories in Figure 5). Accordingly, we observe unchanged performance on LLaVA-Eval (mostly Local tasks) and improvements on subtasks from MME and MathVista (Global and Mixed tasks). To verify the significance of these improvements, we compute McNemar's test p-values on MME and MathVista, as shown in the left table of Table A20. Three out of four training-free sink-to-the-front variants (Intern-VL2.5-4B, Phi3.5-V, and Molmo-7B-D) show statistically significant improvement (p-value < 0.05), with p-values as low as 0.005. Finally, although the training-free variants yield smaller gains than the train-from-scratch

Table A19: Results on the POPE hallucination benchmark. **Bold** indicates the best result per model group. Our method consistently improves over baselines, particularly through dynamic reweighting toward local non-sink tokens.

| Model Name | POPE | | Model Name | POPE |
|---|---|---|---|---|
| *TinyLLaVA 0.5B (Qwen2)* | | | *TinyLLaVA-3B (Qwen2.5)* | |
| Baseline | 86.80 | | Baseline | 85.93 |
| Ours (CoT) | **87.15** | | Ours (CoT) | **86.33** |
| Ours (ReW) | 86.10 | | Ours (ReW) | 86.10 |
| *TinyLLaVA 3B (Phi)* | | | *LLaVA 7B* | |
| Baseline | 84.00 | | Baseline | 87.16 |
| Ours (CoT) | **86.13** | | Ours (CoT) | 86.80 |
| Ours (ReW) | 85.90 | | Ours (ReW) | **87.30** |

Table A20: McNemar's test P-values for performance significance. The left table reports p-values for training-free methods and the right table shows p-values for DIYSink.

| Model | P-value | | Model | P-value |
|---|---|---|---|---|
| InternVL2-4B | 0.0440 | | TinyLLaVA-0.5B (Qwen2) | 0.0061 |
| Phi 3.5 V | 0.0490 | | TinyLLaVA-3B (Phi) | 0.0001 |
| Deepseek-vl-7b-chat | 0.3900 | | TinyLLaVA-3B (Qwen2.5) | 0.0001 |
| MOLMO-7B-D | 0.0050 | | LLaVA-7B | 0.1410 |

models, we present them as a lightweight fallback option, a switch-on mechanism for scenarios where additional data or computation are unavailable.

# D    EXTRA QUALITATIVE EXAMPLES

## D.1    *DIYSink* ON LLAVA-7B

To better understand how the model behaves when sink tokens are emphasized, compared to the baseline that assigns equal weight to sink and non-sink tokens, we present qualitative examples using *DIYSink* (ReW) on LLaVA-7B. These examples, shown in Fig. A7, Fig. A8, and Fig. A9, illustrate the model's reasoning process on commonsense reasoning, counting, and arithmetic tasks. We observe that reweighting the sink tokens enables the model to produce more coherent and accurate reasoning, particularly for questions that require global understanding and higher-level inference.

## D.2    WORD MAPPING RESULTS

We provide some qualiatitve examples in Fig. A11 and Fig. A12 of the word mapping result as illustrated in Sec. 3.2. Interestingly, we observed accurate mapping of patches with open-form words such as the "cup" and "water-other" in the second figure of Fig. A11, as well as the "window-blind" in the first figure of Fig. A12 and the "cell phone" in the second figure of Fig. A12

# E    MODEL LIMITATIONS AND SOCIAL IMPACT

## E.1    MODEL LIMITATIONS AND FUTURE DIRECTIONS

While our training-free repositioning strategy generally enhances the VQA capabilities of recent LVLMs, our proposed *DIYSink* framework–though yielding more substantial improvements–requires access to training data and certain amount of computational resources, as it involves training the model from scratch.

**Commonsense Reasoning**

**?** Does the vehicle in the picture usually run faster than the car?

**LLaVA-7B + DIYSink** ✓

**A:** No, the vehicle in the picture, which is a bicycle, does not usually run faster than a car. Bicycles are human-powered and rely on the rider's physical strength and stamina to propel them forward. Cars, on the other hand, are motorized vehicles that use engines to generate power and speed. Bicycles are generally slower than cars, ...

**LLaVA-7B** ✗

**A:** Yes, the bicycle in the picture usually runs faster than the car. Bicycles are human-powered vehicles that rely on the rider's physical strength and endurance to propel them forward. This makes them capable of reaching higher speeds than cars...

**Sink vs. Non-sink tokens weight [1.14, 0.86]**

Figure A7: **Qualitative comparison** of LLaVA 7B + *DIYSink* and LLaVA 7B on the Commonsense Reasoning task.

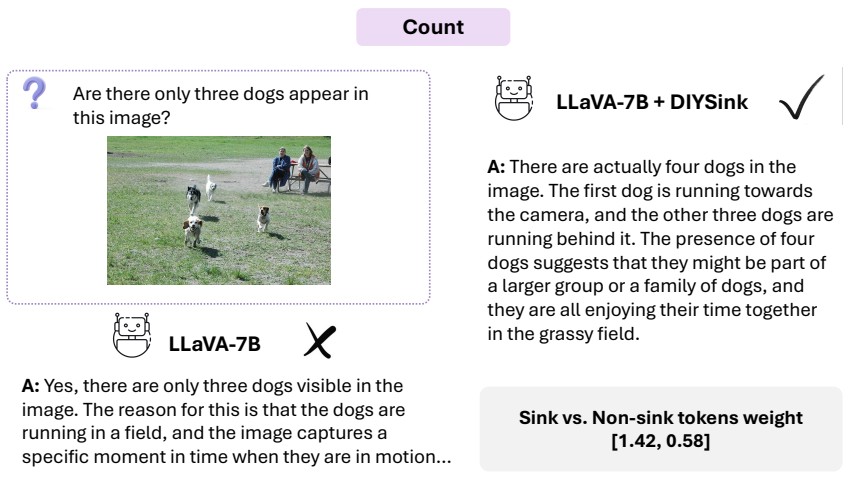

**Count**

**?** Are there only three dogs appear in this image?

**LLaVA-7B + DIYSink** ✓

**A:** There are actually four dogs in the image. The first dog is running towards the camera, and the other three dogs are running behind it. The presence of four dogs suggests that they might be part of a larger group or a family of dogs, and they are all enjoying their time together in the grassy field.

**LLaVA-7B** ✗

**A:** Yes, there are only three dogs visible in the image. The reason for this is that the dogs are running in a field, and the image captures a specific moment in time when they are in motion...

**Sink vs. Non-sink tokens weight [1.42, 0.58]**

Figure A8: **Qualitative comparison** of LLaVA 7B + *DIYSink* and LLaVA 7B on the counting task.

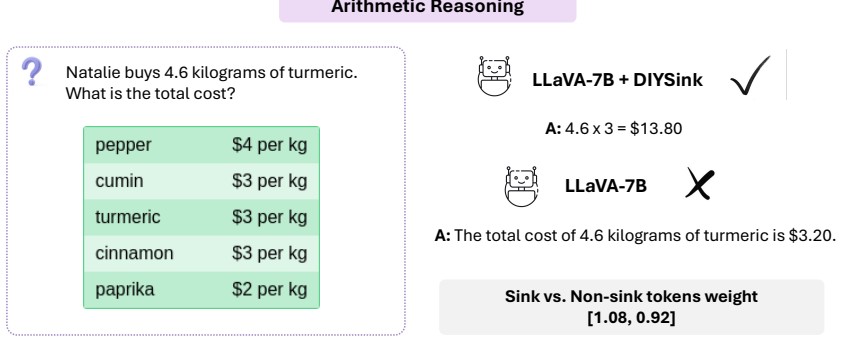

**Arithmetic Reasoning**

**?** Natalie buys 4.6 kilograms of turmeric. What is the total cost?

| pepper | $4 per kg |
| cumin | $3 per kg |
| turmeric | $3 per kg |
| cinnamon | $3 per kg |
| paprika | $2 per kg |

**LLaVA-7B + DIYSink** ✓

**A:** 4.6 x 3 = $13.80

**LLaVA-7B** ✗

**A:** The total cost of 4.6 kilograms of turmeric is $3.20.

**Sink vs. Non-sink tokens weight [1.08, 0.92]**

Figure A9: **Qualitative comparison** of LLaVA 7B + *DIYSink* and LLaVA 7B on the arithmetic reasoning task.

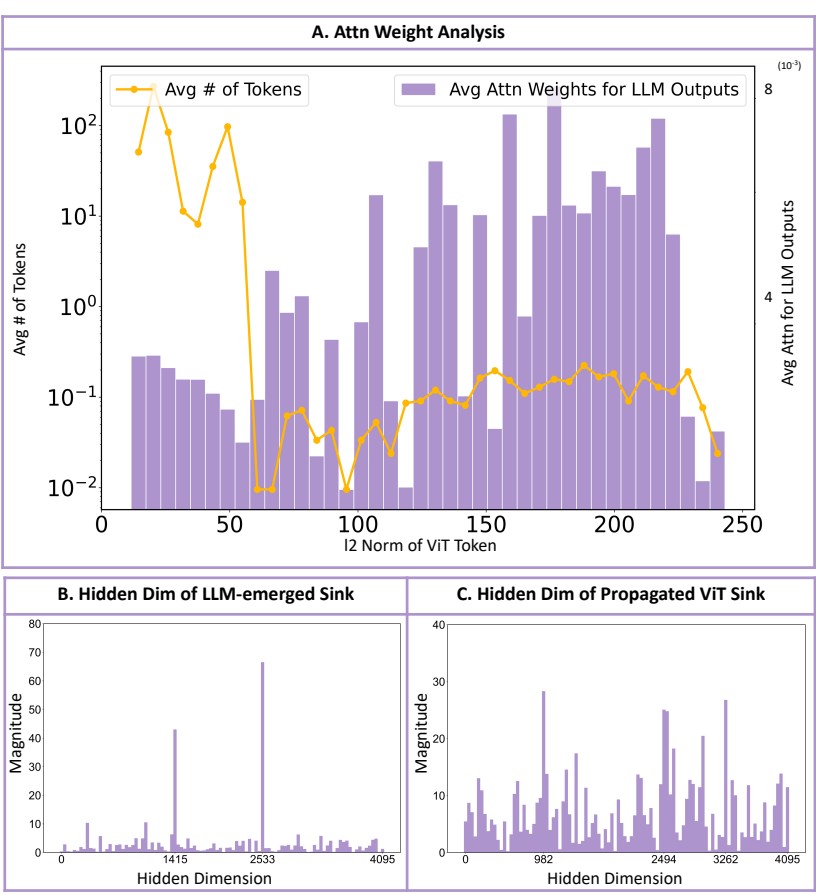

Figure A10: Attention to ViT sink tokens and sink dimensions of ViT and LLM sinks in LLaVA-v1.5-7B. Enlarged figure 3.

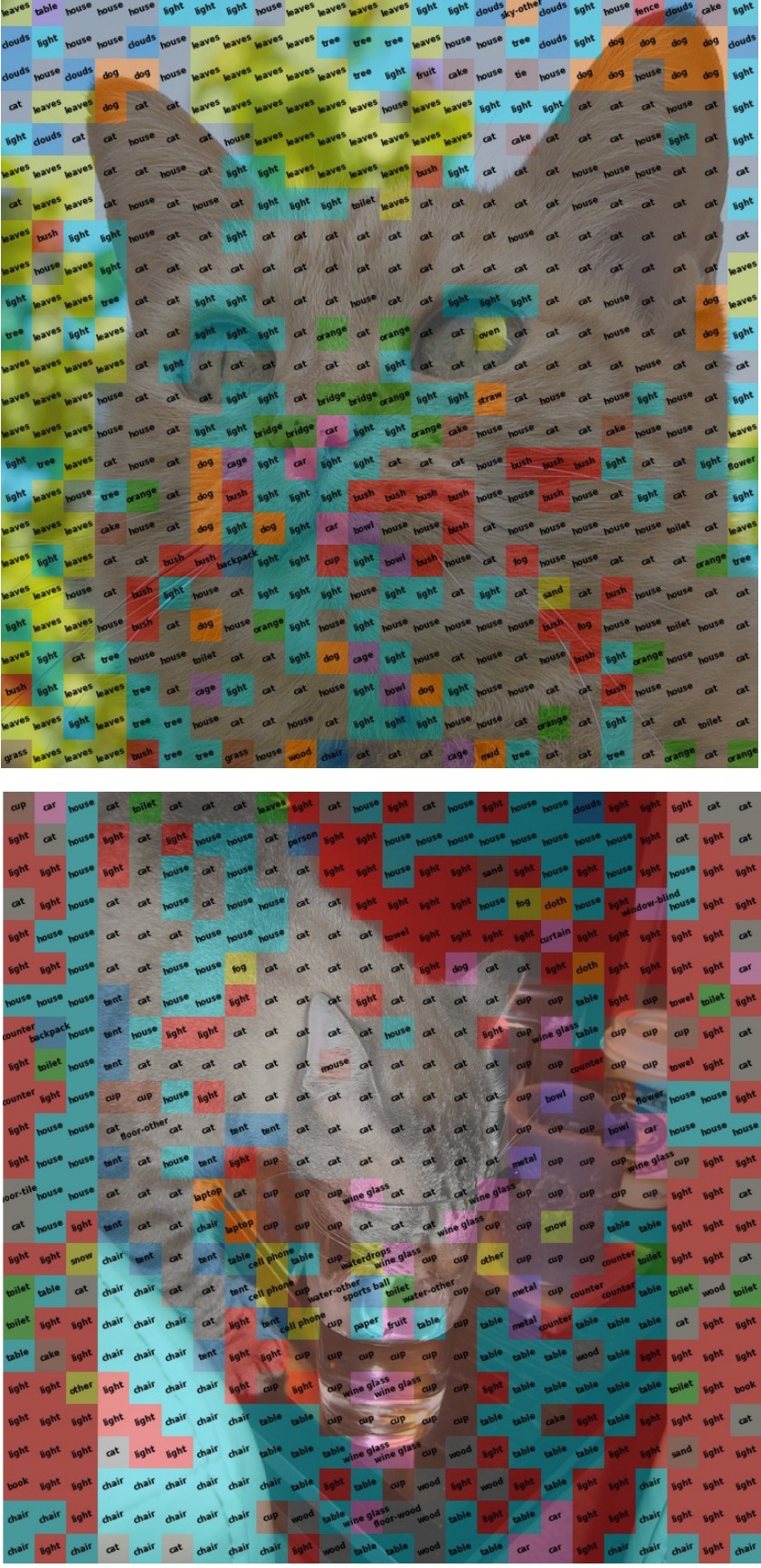

Figure A11: Qualitative examples of word mapping.

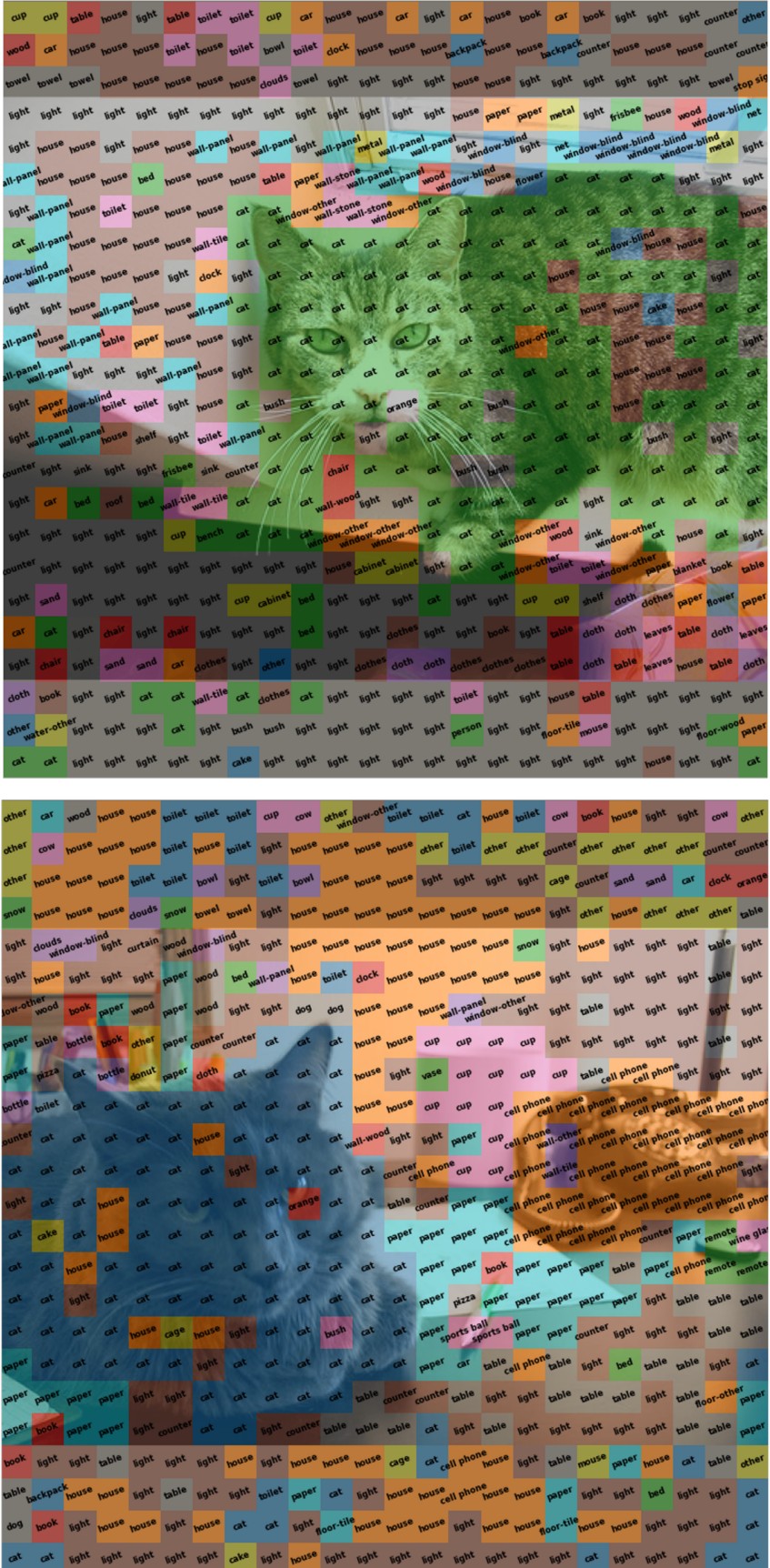

Figure A12: Qualitative examples of word mapping.

Additionally, this work primarily focuses on the image and language modalities. Extending *DIYSink* to other modalities such as video and audio remains an open area for future research and exploration.

### E.2 SOCIAL IMPACT

In this work, we demonstrate that ViT sink tokens encode high-level visual context and can be effectively utilized by LVLMs to solve tasks requiring global understanding and reasoning. Building on this insight, we propose two approaches to enhance the use of ViT sinks in existing LVLMs: (1) a training-free repositioning strategy that prioritizes ViT sink tokens in the input sequence, and (2) *DIYSink*, a training-based framework incorporating dual MLP projectors and task-aware selection modules. Our methods yield significant improvements over the baseline across a range of VQA tasks.

While our research focuses on improving model performance, it is important to acknowledge potential misuse. In particular, large vision-language models (VLMs) could generate harmful visual or textual content if deployed irresponsibly. Although this risk is not specific to our methods, it highlights the need for further research into safety and ethical considerations in multimodal AI systems.

## F   LLM USAGE

In this paper, we utilized LLMs as a writing assistant. The main usage of LLM was for proofreading, enhancing text fluency, and checking grammar correctness. All the scientific contributions, findings, experimental design, and framework design are the original work of the authors.

