# OpenReview forum: "To Sink or Not to Sink: Visual Information Pathways in Large Vision-Language Models"
_ICLR.cc/2026/Conference — ICLR 2026 Poster_

### Official Review · Reviewer_4Rra · 2025-10-14

**Soundness:** 3
**Presentation:** 3
**Contribution:** 4
**Rating:** 8
**Confidence:** 4

**Summary:**

This paper studies the effect of (ViT) attention sink in vision language models. It finds that a few high norm ViT tokens act as visual sinks that receive disproportionate attention in the LLM and carry coarse scene information. Building on this, the paper proposes a training-free token reordering and a trainable MLP connector with dynamic routing to improve various downstream visual reasoning tasks. The findings about that the sink tokens are beneficial for tasks with low visual complexity and global semantics but may degrade performance
on tasks that demand localized, detail-rich visual processing is useful.

**Strengths:**

1 The paper is well-written and easy to follow.

2 The research topic is timely and interesting. The paper shows a clear empirical pattern linking high norm visual tokens to downstream concept semantics and reasoning capacity.

3 The proposed strategies that can be adopted with low engineering overhead.

4 The use of relevance map and decoding to world distribution to study the propagation of ViT attention sink to VLM is a sound approach.

5 The finding that the ViT sinks in VLMs encode vital high-level visual context and is actually useful for visual reasoning task is a novel finding and good to know for the community.

**Weaknesses:**

1. While the link between attention sink and visual semantics are evident. Why making the sink token to the front/end would make visual reasoning capacity stronger/weaker in the training-free setting, supposing the connector still struggles to project the re-ordered sink/non-sink tokens to the same embedding?
2. Currently all experiments are conducted up to 7b models, have the authors consider to test larger models like to see if the observed patterns generalize?

- Figure A9/A10's captions are "Enter Caption"
- Font size in Figure 4/5 too small

**Questions:**

1. How does RoPE handle the sink token reordering at inference time, as positional embeddings of the sink token are not the same as what the VLMs see during training (i.e. without reordering)? Does moving sinks to the front/back introduce unintended positional biases?

---

> ### Author Response · Authors · 2025-11-25
> **Official Comment by Authors (1/2)**
>
> We sincerely appreciate the thoughtful feedback from Reviewers qH3u, UaWB, yLnB, ZDRY, and 4Rra, as well as their constructive suggestions on how to improve the paper. We are deeply encouraged that reviewers found our findings novel and interesting (ZDRY, qH3u, 4Rra) and valuable to the community (qH3u, 4Rra). Reviewers also noted that our paper is well-structured with clear motivation (qH3u), offers thorough analysis (yLnB, ZDRY), provides sufficient qualitative and quantitative evidence to support our findings (yLnB), and presents a sound approach and application (qH3u, 4Rra); our proposed methods incur low engineering overhead (4Rra) and are intuitive, simple yet effective, practical, and easy to integrate into existing LVLMs (ZDRY, yLnB, UaWB); our paper provide comprehensive experiments across diverse models and benchmarks (qH3u), showing consistent performance gains (yLnB), and recognized the strong potential for visual token compression and efficient inference suggested by our work (UaWB). We greatly appreciate the time reviewers spent evaluating our paper and offering insightful comments. We address each concern in detail in our responses, and we would be happy to clarify any additional questions during the discussion phase.
>
> > Q1: Why making the sink token to the front/end would make visual reasoning capacity stronger/weaker in the training-free setting, supposing the connector still struggles to project the re-ordered sink/non-sink tokens to the same embedding?
>
> **A: Summary/TL;DR: Sink-to-front works because causal attention in LLM amplifies high-level sink information, even with an imperfect connector; our dual-MLP improves this by processing two token types individually.**
>
> Although the connector MLP in the training-free setting is not optimized to jointly handle two distinct types of visual tokens, each token type is still partially preserved. In this setting, moving sink tokens to the front remains beneficial because, under the LLM’s causal attention structure, more subsequent tokens can attend to and incorporate the coarse-grained, high-level information encoded by the sink tokens.
>
> In contrast, our dual-MLP design more effectively preserves the semantic information in sink and non-sink tokens by projecting each type through its own dedicated pathway. This enables the model to fully leverage both coarse-level and fine-level visual information, leading to larger performance gains than those achieved by the training-free approach. We also investigate the reason for repositioning to the front instead of the end. Please refer to Q7 of Reviewer ZDRY for more details.
>
> > Q2: Can the authors test larger models to see if the observed patterns generalize?
>
> **A: Summary/TL;DR: We demonstrate that our training-free sink-to-the-front approach yields performance improvements on Gemma-3-12B-it.**
>
> We thank the reviewer for their constructive feedback and the suggestion to evaluate our method on larger or more recent architectures. Due to current hardware constraints, we utilized the Gemma-3-12B-it model for additional validation, comparing the baseline against our proposed training-free “sink-to-front” method. We use the official codebase on huggingface to perform inference. The results confirm the effectiveness and generalizability of our approach on larger model. Specifically, on the MME benchmark, our method yields a significant improvement, boosting the Cognition from 535.36 to 563.21 and raising the overall score to 1740.11. This demonstrates enhanced reasoning capability. Furthermore, on general VQA tasks and MathVista, the method improves overall performance, confirming that leveraging attention sink is an effective strategy for improving model performance without degrading foundational visual understanding.
>
> | Method | GQA | SQA-image | TextVQA | MMMU-val | MME |  |  |  |  | MathVista |  |  |  |
> |---|---|---|---|---|---|---|---|---|---|---|---|---|---|
> | | | | | | ALL | Cognition | ComRea | TextTrans | CodeRea | ALL | ALG | GEO | LOG |
> | **Gemma3-it** | 45.24 | **82.55** | 70.23 | 47.00 | 1706.30 | 535.36 | 127.86 | 112.50 | 162.50 | 40.80 | 48.4 | 48.95 | **13.51** |
> | **Gemma3-it sink-to-front** | **45.41** | 82.50 | **70.30** | **48.00** | **1740.11** | **563.214** | **130.71** | **130.00** | **170.00** | **41.00** | **50.89** | **51.46** | 8.11 |

---

> ### Author Response · Authors · 2025-11-25
> **Official Comment by Authors (2/2)**
>
> > Q3: How does RoPE handle the sink token reordering at inference time, as positional embeddings of the sink token are not the same as what the VLMs see during training (i.e. without reordering)? Does moving sinks to the front/back introduce unintended positional biases?
>
> **A: Summary/TL;DR: For training-free methods, we "jointly reposition" sink tokens with their original positional embeddings. Therefore, when generating outputs, the model observes the same spatial positioning for sink tokens as prior to reordering.**
>
> We thank the reviewer for this insightful question regarding RoPE and token reordering. We clarify that the training-free inference process is consistent with the model’s initial training; as described in Line 322, we do not simply move the token; we "jointly reposition these sink tokens, along with their associated positional embeddings, to the beginning of the visual sequence" during inference. Mathematically, let $R_{\Theta, m}$ denote the RoPE rotation matrix at position $m$. The attention score relies on the relative distance between query position $m$ and key position $n$: specifically, if we rotate the inputs such that $q = R_{\Theta, m} x_q$ and $k = R_{\Theta, n} x_k$, their inner product $q^T k$ mathematically simplifies to $x_q^T R_{\Theta, n-m} x_k$. Since we reposition the tokens with their initial positional embeddings during inference, the assigned position index $m_{sink}$ is invariant. This ensures that when generating output, the relative distance $n_{output} - m_{sink}$ remains consistent, while allowing $m_{sink}$ to be perceived by subsequent visual tokens. Therefore, the reordering does not introduce unintended positional biases in the attention weights; rather, it allows us to intentionally position the sink token within the causal mask to maximize its visibility.
>
> > Typos and missing captions for figures in the appendix
>
> **A**: We thanks reviewer for pointing out the typos and missing captions. We had fixed them in the revision.

---

### Official Review · Reviewer_ZDRY · 2025-10-27

**Soundness:** 2
**Presentation:** 2
**Contribution:** 1
**Rating:** 2
**Confidence:** 4

**Summary:**

This paper investigates the role of attention sink tokens originating from the Vision Transformer (ViT) in Large Vision-Language Models (LVLMs). Unlike prior work that focuses on eliminating or suppressing attention sinks as harmful, this work finds that a subset of ViT sinks propagate into the LLM and encode high-level semantic information. The authors find that ViT sinks help on global reasoning tasks but can hurt performance on fine-grained and local visual tasks. Based on this insight, they propose two methods to leverage sink tokens: a training-free “sink-to-the-front” token reordering strategy, and a training-based DIYSink framework with dynamic token selection. The experiments across multiple LVLM architectures show consistent improvements, especially on MathVista, MME, and reasoning benchmarks.

**Strengths:**

- **Novel insights**: The paper questions the common belief that visual attention sinks are harmful. Instead, it shows that ViTs’ sink tokens can carry useful high-level information, which can help LVLMs with global visual reasoning.
- **Thorough empirical analysis**: The authors study the phenomenon from several angles, such as attention patterns and feature norms. The experiments are thorough and support the main claims well.
- **Practical proposed methods**: “The proposed “sink-to-the-front” approach is training-free, simple to use, and still leads to improvements.

**Weaknesses:**

- **Limited theoretical explanation**: The paper offers little insight into the main claim—that ViT sink tokens encode global semantics. The explanation remains largely observational, with no theoretical grounding or analytical model to support why this behavior arises. More intuition or theoretical explanation would strengthen the contributions.
- **Only marginal improvements**: Although the results show gains, many improvements reported in Tables 1 and 2 are relatively small—often below 1% and sometimes within the range of natural evaluation variance. Statistical tests or error bars would help validate that the gains are meaningful rather than just noise.
- **Limited evidence on larger models**: The experiments are run on models up to 7B parameters. It is unclear if the improvements would remain for much larger models (> 7B), which may behave differently. Any experiment or discussion on this topic is valuable.
- **Generalization beyond ViTs**: It is unclear whether the same sink behavior holds —and whether the method would work— for CNN-based or Mamba-based vision encoders, or for architectures that do not use MLP connectors.
- **Incremental contribution**: Overall, the paper provides an interesting empirical observation but lacks sufficient novelty, depth, or quantitative impact to justify acceptance.

**Questions:**

1. Would the same sink behavior appear in models using non-ViT vision encoders? Or when using connector architectures other than an MLP, such as cross-attention or Q-Former?
2. For DIYSink, how sensitive is the performance to incorrect task classification (e.g. when a “global” task is misclassified as “local”)?
3. Why does placing sink tokens at the front work better than at the end of the token sequence? Could relative positional encodings influence this outcome?
4. Have you tried the "sink-to-the-front" approach for other VLMs, such as LLaVA models: TinyLLaVA-3B or LLaVA-7B?
5. Could you also address the points mentioned in the weaknesses list?

---

> ### Author Response · Authors · 2025-11-25
> **Official Comment by Authors (1/5)**
>
> We sincerely appreciate the thoughtful feedback from Reviewers qH3u, UaWB, yLnB, ZDRY, and 4Rra, as well as their constructive suggestions on how to improve the paper. We are deeply encouraged that reviewers found our findings novel and interesting (ZDRY, qH3u, 4Rra) and valuable to the community (qH3u, 4Rra). Reviewers also noted that our paper is well-structured with clear motivation (qH3u), offers thorough analysis (yLnB, ZDRY), provides sufficient qualitative and quantitative evidence to support our findings (yLnB), and presents a sound approach and application (qH3u, 4Rra); our proposed methods incur low engineering overhead (4Rra) and are intuitive, simple yet effective, practical, and easy to integrate into existing LVLMs (ZDRY, yLnB, UaWB); our paper provide comprehensive experiments across diverse models and benchmarks (qH3u), showing consistent performance gains (yLnB), and recognized the strong potential for visual token compression and efficient inference suggested by our work (UaWB). We greatly appreciate the time reviewers spent evaluating our paper and offering insightful comments. We address each concern in detail in our responses, and we would be happy to clarify any additional questions during the discussion phase.
>
> > Q1: This paper lacks a theoretical explanation. The explanation remains largely observational, with no theoretical grounding or analytical model to support why this behavior arises. More intuition or theoretical explanation would strengthen the contributions.
>
> **A: Summary/TL;DR: Because the scale and complexity of LVLMs make rigorous theoretic analysis difficult, our study, as are others, relies on empirical evidence to illuminate their internal mechanisms. However, we do rely on observations from other works and our own intuition for designs and hypotheses that we then implement and empirically verify. Below, we provide a deeper discussion considering the intrinsic properties and architectures of LVLMs.**
>
> Based on the previous NLP literature [1,2,3], sinks arise from a combination of architectural biases and the softmax normalization in self‐attention. First, early layers produce some patch with a small initial advantage e.g. slightly higher norm in certain hidden dimensions. These are usually low semantic tokens as the model finds them semantically-free to sacrifice [2,4]. As the layer accumulates and because softmax keeps all allowed edges nonzero, these tokens absorb information from the later tokens and attract attention as models propagate through layers [1]. This explains why over middle layers these “massive tokens” emerge [4] and why models with more layers and longer pretraining context would have more prominent sinks [2]. In the meantime, RoPE [5] induces positional bias such that the dot-product (and attention) for tokens that are far apart gradually vanishes or even turns negative. That breaks the attention map into smaller groups. Inside any group, information can still drift toward tokens with initial advantage, resulting in multiple sink tokens in transformer architecture.
>
> Since sink tokens from ViT and LLM emerge during their own pretraining, they activate different hidden dimensions of the sinks. Also because sink tokens absorb information from other tokens as models propagate, they include aggregated coarse-grained information of the image/sentence. This explains our main observations in Section 3. Since sink tokens include summarization information and they have different hidden dimension value distribution as non-sink tokens, we require dual-mlp to project different tokens appropriately and then we rely on reweighting to highlight sinks when tasks require high-level understanding and de-emphasize sink tokens when tasks require local understanding.
>
> ---
> **Reference**
>
> [1] Wang, Z., Zhang, H., Li, X., Huang, K.H., Han, C., Ji, S., Kakade, S.M., Peng, H. and Ji, H., 2024. Eliminating position bias of language models: A mechanistic approach. arXiv preprint arXiv:2407.01100.
>
> [2] Barbero, F., Arroyo, A., Gu, X., Perivolaropoulos, C., Bronstein, M., Veličković, P. and Pascanu, R., 2025. Why do LLMs attend to the first token?. arXiv preprint arXiv:2504.02732.
>
> [3] Gu, X., Pang, T., Du, C., Liu, Q., Zhang, F., Du, C., Wang, Y. and Lin, M., 2024. When attention sink emerges in language models: An empirical view. arXiv preprint arXiv:2410.10781.
>
> [4] Sun, M., Chen, X., Kolter, J.Z. and Liu, Z., 2024. Massive activations in large language models. arXiv preprint arXiv:2402.17762.
>
> [5]  Su, J., Lu, Y., Pan, S., Murtadha, A., Wen, B. and Roformer, Y.L., 2023. Enhanced transformer with rotary position embedding., 2021. DOI: https://doi. org/10.1016/j. neucom.

---

> ### Author Response · Authors · 2025-11-25
> **Official Comment by Authors (2/5)**
>
> > Q2: The reported improvements are marginal. Statistical tests or error bars are requested to validate significance.
>
> **A: Summary / TL;DR: Three of the four train-from-scratch variants show significant improvement (p-value < 0.05) and one show moderate significance. Three out of four of our training-free sink-to-the-front variants also show significant improvement (p-value < 0.05) on tasks that require sink tokens.**
>
> We appreciate the reviewer’s comment and are pleased to provide additional validation. We perform McNemar’s test [1,2] to compare the performance of both the training-free and train-from-scratch methods against the baseline.
> For the train-from-scratch methods, as shown in Table 1, we compute McNemar’s test p-values over all benchmarks for DIYSink (Rew) vs. baseline while making sure equal weight between benchmarks and obtain statistically significant results for three out of four models. LLava 7B’ s p-value is not as significant but the p-value = 0.14 is still showing some significance.
>
> For the training-free methods, as discussed in Section 3, sink tokens benefit the model primarily on tasks requiring high-level, abstract reasoning (the Global and Mixed categories in Section 3.3 and Figure 5). Accordingly, we observe unchanged performance on LLaVA-Eval (mostly Local tasks) and improvements on subtasks from MME and MathVista (Global and Mixed tasks). To verify the significance of these improvements, we compute McNemar’s test p-values on MME and MathVista, as shown in Table 2 below. Three out of four training-free sink-to-the-front variants (Intern-VL2.5-4B, Phi3.5-V, and Molmo-7B-D) show statistically significant improvement (p-value < 0.05), with p-values as low as 0.005. We will include the detailed procedures and results in our supplemental material. Finally, although the training-free variants yield smaller gains than the train-from-scratch models, we present them as a lightweight fallback option, a switch-on mechanism for scenarios where additional data or computation are unavailable.
>
> Note: All p-values are rounded to four decimal places.
>
> **Table 1**
> | Model | McNemar's test P-value |
> | :--- | :--- |
> | TinyLLaVA-0.5B (Qwen2) |   0.0061 |
> | TinyLLaVA-3B (Phi) |   0.0001 |
> | TinyLLaVA-3B (Qwen2.5) |   0.0001 |
> | LLaVA-7B|   0.1410 |
>
> **Table 2**
> | Model | McNemar's test P-value |
> | :--- | :--- |
> | InternVL2-4B | 0.0440 |
> | Phi 3.5 V | 0.0490 |
> | Deepseek-vl-7b-chat | 0.3900 |
> | MOLMO-7B-D | 0.0050 |
>
> > Q3: Limited evidence on larger models: The experiments are run on models up to 7B parameters. It is unclear if the improvements would remain for much larger models (> 7B), which may behave differently. Any experiment or discussion on this topic is valuable.
>
> **A: Summary/TL;DR: We demonstrate that our training-free sink-to-the-front approach yields performance improvements on Gemma-3-12B-it.**
>
> We thank the reviewer for their constructive feedback and the suggestion to evaluate our method on larger or more recent architectures. Considering hardware constraints, we utilized the Gemma-3-12B-it model for additional validation, comparing the baseline against our proposed training-free “sink-to-front” method. We use the official codebase on huggingface to perform inference. The results confirm the effectiveness and generalizability of our approach on larger models. Specifically, on the MME benchmark, our method yields a significant improvement, boosting the Cognition from 535.36 to 563.21 and raising the overall score to 1740.11. This demonstrates enhanced reasoning capability. Furthermore, on general VQA tasks and MathVista, the method improves overall performance, confirming that leveraging attention sink is an effective strategy for improving model performance without degrading foundational visual understanding.
>
> | Method | GQA | SQA-image | TextVQA | MMMU-val | MME |  |  |  |  | MathVista |  |  |  |
> |---|---|---|---|---|---|---|---|---|---|---|---|---|---|
> | | | | | | ALL | Cognition | ComRea | TextTrans | CodeRea | ALL | ALG | GEO | LOG |
> | **Gemma3-it** | 45.24 | **82.55** | 70.23 | 47.00 | 1706.30 | 535.36 | 127.86 | 112.50 | 162.50 | 40.80 | 48.4 | 48.95 | **13.51** |
> | **Sink-to-the-front (Ours)** | **45.41** | 82.50 | **70.30** | **48.00** | **1740.11** | **563.214** | **130.71** | **130.00** | **170.00** | **41.00** | **50.89** | **51.46** | 8.11 |
>
> ---
> **Reference**
>
> [1] McNemar, Q., 1947. Note on the sampling error of the difference between correlated proportions or percentages. Psychometrika, 12(2), pp.153-157.
>
> [2] Raschka, S., 2018. Model evaluation, model selection, and algorithm selection in machine learning. arXiv preprint arXiv:1811.12808.

---

> ### Author Response · Authors · 2025-11-25
> **Official Comment by Authors (3/5)**
>
> ### **Note: The references are placed in the last response.**
> > Q4: Generalization beyond ViTs: It is unclear whether the same sink behavior holds for CNN-based or Mamba-based vision encoders. Or for other types of connectors other than an MLP, such as cross-attention or Q-Former?
>
> **A: Summary/TL;DR: For visual encoder, we have not found any LVLMs that use a Mamba- or CNN-based visual encoder, although Mamba does appear in LVLMs as a connector or as the language model (e.g. Meteor [1]). We further implement our training-free sink-to-the-front method and show improved performance on Meteor [1].**
>
> **For the connector, we replaced TinyLLaVA-Qwen2-0.5B’s MLP connector with either a Q-former or a Perceiver Resampler and trained these variants from scratch with DIYSink. Both alternatives yielded consistent improvements across benchmarks, demonstrating that DIYSink generalizes well across different connector architectures.**
>
> | Method | GQA | SQA-image | TextVQA | MMMU-val | MME |  |  |  |  | MathVista |  |  |  |
> |---|---|---|---|---|---|---|---|---|---|---|---|---|---|
> | | | | | | ALL | ComRea | NumCal | TextTrans | CodeRea | ALL | ARI | GEO | LOG |
> | **Meteor Baseline** | 62.54 | 94.29 | 67.01 | 39.00 | 2188.75 | 140.71 | 105.00 | 160.00 | 102.50 | 52.80 | 52.97 | 42.68 | 13.51 |
> | **Sink-to-the-front (Ours)** | **62.57** | **94.65** | **67.07** | **39.10** | **2198.47** | **141.42** | **105.00** | **160.00** | **110.00** | **53.00** | **53.54** | **42.68** | **13.51** |
>
> We appreciate the reviewer’s concern and are happy to provide additional information. To our best knowledge, in the survey of prior work [3–12], no recent LVLM employs a CNN-based or Mamba-based model as the visual encoder; ViT-based encoders such as CLIP and SigLIP remain the dominant architecture. Nevertheless, there have been some explorations of using Mamba as a connector or as the language model. To test how our method performs on such models, we applied our training-free method to Meteor [1], a LVLM using Mamba as connector and LLM, and observed performance improvements, as shown in the table above.
> Moreover, although no existing LVLMs employ a Mamba-based ViT, prior work [2] has shown that sink tokens also emerge in Mamba based visual encoder. This suggests that DIYSink has the potential to naturally extend to future LVLMs that adopt Mamba-style visual encoders, further broadening its applicability.
>
> To further investigate the impact of connector architectures, we studied two cross-attention-based connectors: the Q-former and the Perceiver Resampler [13]. We implemented these connectors within the TinyLLaVA-Qwen2, 0.5B model architecture, replacing the standard MLP connector, and trained the models from scratch to establish baselines. In both architectures, the connector takes 728 visual tokens as input and outputs a compressed representation of 128 tokens via cross-attention. We then applied our DIYSink method. For a fair comparison with the baseline, we maintained the total output token count at 128 by setting the sink token connector (query/latent) size to 2 and the non-sink token size to 126. Following dual-connector training and chain-of-thought inference, the results demonstrate that both the Q-former and the Perceiver Resampler achieve consistent improvements on the MME and MathVista benchmarks. Overall scores across general benchmarks also increased for both models. This strongly indicates that leveraging ViT sink tokens is crucial for enhancing the reasoning capabilities of Large Vision-Language Models (LVLMs), even when employing cross-attention-based connectors.
>
> | Method | GQA | SQA-image | TextVQA | MMMU-val | MME |  |  |  |  | MathVista |  |  |  |
> |---|---|---|---|---|---|---|---|---|---|---|---|---|---|
> | | | | | | ALL | Cognition | ComRea | TextTrans | CodeRea | ALL | ALG | GEO | LOG |
> | **Tinyllava 0.5B (Qwen) - qformer** | 47.75 | 55.78 | 33.53 | **31.00** | 1160.33 | 221.43 | 71.43 | 47.50 | **50.00** | 23.30 | 24.56 | 23.01 | 13.51 |
> | **Tinyllava 0.5B (Qwen) - qformer (dynamic cot)** | **48.01** | **56.62** | **33.93** | 30.00 | **1173.84** | **257.86** | **77.86** | **75.00** | 47.50 | **23.80** | **25.98** | **25.10** | **13.51** |
> | **Tinyllava 0.5B (Qwen) - resampler** | 53.65 | **61.18** | **45.30** | 32.00 | 1403.31 | 234.29 | 84.29 | 50.00 | 50.00 | 23.60 | 21.00 | 20.08 | **24.32** |
> | **Tinyllava 0.5B (Qwen) - resampler (dynamic cot)** | **53.75** | 60.93 | 45.12 | **32.00** | **1427.11** | **237.50** | **85.00** | **50.00** | **50.00** | **24.00** | **22.42** | **22.18** | 21.62 |

---

> ### Author Response · Authors · 2025-11-25
> **Official Comment by Authors (4/5)**
>
> > Q5: The contribution of this paper is incremental and lacks sufficient novelty, depth, or quantitative impact to justify acceptance.
>
> We thank the reviewer for their feedback regarding our contributions and would like to clarify them further. First, we analyze the fundamental behavior of ViT sinks, ranging from token propagation and hidden dimension breakdown to their specific role in LLM reasoning. Second, we propose two strategies (one training-free and one training-from-scratch) that enable LVLMs to dynamically leverage sink tokens when reasoning is required. To the best of our knowledge, this is the first work to investigate the impact of ViT sink tokens within LLMs, a topic of fundamental importance. Extensive experiments on 4 closed-source and 4 open-source models across 6 benchmarks demonstrate the effectiveness of our approach.
>
> > Q6: For DIYSink, how sensitive is the performance to incorrect task classification (e.g. when a “global” task is misclassified as “local”)?
>
> **A: Summary/TL;DR: Misclassifying global and local tasks leads to substantial performance drops. However, our reweighting modules achieve very low misclassification rates. As a result, the dynamic selection module in DIYSink consistently improves downstream performance over the baseline by correctly leveraging the appropriate token type.**
>
> As shown in the table above, sink and non-sink tokens behave very differently on global versus local tasks. When all local questions are mistakenly treated as global, as in the Ours (sink) setting, where only sink tokens are used, the performance on local tasks drops sharply, often falling well below the baseline. Conversely, in the Ours (nonsink) setting, using only non-sink tokens causes substantial degradation on global tasks when they are misclassified as local.
>
> However, leveraging ground truth labeling from human evaluation in Q2 of Reviewer UaWB, we found the misclassification rate to be only 23% and 18% for the 0.5B and 3B model respectively. Specifically, we collected human-labeled task classifications (i.e., global, local, mix) on 64 VQA samples from 6 benchmarks and compared them with the predictions from LVLMs. LVLMs successfully classify most cases. With the correct classification, we observe consistent improvement in downstream tasks with our dynamic selection module, as shown in the table below (DIYSink CoT and DITSink ReW). This demonstrates that DIYSink can adaptively choose the suitable token type for each task and approximate maximum performance, effectively leveraging the strengths of both sink and non-sink tokens. Note that when performing CoT or reweighting in DIYSink, GPT is not involved in the decision-making; GPT classification is only used for the analysis in Section 3.
>
> **Note**: Local represent tasks that non sink tokens work well and Global represent tasks that sink tokens work well. **We bold the numbers where DIYSink surpass baseline as we are comparing with baseline here.**
> | | Local | | | Global |  |  |
> | - | --- | --- | --- | --- |--- | --- |
> | Methods | GQA | SQA-image | TextVQA | MME Cognition | MathVista SciRea | MathVista NUM |
> | **Tinyllava 0.5B (Qwen)** | 57.98 | 57.76 | 47.32 | 207.14 | 36.89 | 11.11 |
> | Ours (sink) | 39.53 | 53.94 | 30.10 | 270.71 | 37.70 | 17.36 |
> | Ours (nonsink) | 58.07  | 60.09 | 48.28 | 220.00 | 36.89 | 11.11 |
> | DIYSink (CoT) | **58.14** | **61.08** | 46.14 | **277.50** | **37.70** | **11.81** |
> | DIYSink (ReW) | 57.75 | **60.24** | **47.42** | **229.64** | **38.52** | **14.58** |
> | | | | | | | |
> | **Tinyllava 3B (Phi)** | 51.09 | 68.47 | 41.15 | 261.79 | 45.90 | 13.89 |
> | Ours (sink) | 39.13 | 65.79 | 35.92 | 315.71 | 45.90 | 22.22 |
> | Ours (nonsink) | 60.02 | 70.25 | 47.51 | 269.64 | 43.44 | 19.44 |
> | DIYSink (CoT) | **59.99** | **68.57** | **46.22** | **280.71** | **47.54** | **13.89** |
> | DIYSink (ReW) | **59.79** | **70.50** | **50.77** | **265.71** | 42.62 | **19.44** |

---

> ### Author Response · Authors · 2025-11-25
> **Official Comment by Authors (5/5)**
>
> ### **Note: The references are placed in the last response.**
> > Q7: Why does placing sink tokens at the front work better than at the end of the token sequence? Could relative positional encodings influence this outcome?
>
> **A: Summary/TL;DR: Because the bias from causal attention in LLMs outweighs the recency bias of Rotary Positional Embeddings (ROPE), positioning sink tokens at the front yields better performance.**
>
> We thank the reviewer for this insightful question. Early in this project, we explored this question and included the analysis result in Section C.1 of the supplementary as mentioned in L454 of the main paper. We reiterate the experiment results here:
> The primary motivation for repositioning sink tokens is to amplify their impact during output generation, which we can achieve by leveraging two positional biases in LLMs:
>
> *Causal attention bias*: In autoregressive Transformers, each token attends only to its predecessors, amplifying the influence of early tokens because information flows forward but not backward. By moving sink tokens, together with their original positional embeddings, to the front of the sequence, we exploit this bias.
>
> *RoPE recency bias* [14-16]: Rotary Position Embeddings introduce a mild “recency” effect: attention weights decay as the relative distance between query and key increases, causing the model to favor tokens nearer the end of the sequence. By repositioning tokens to the end and reassigning their positional embeddings, we can also leverage this effect.
>
> Although both biases shape attention, our empirical study (see table above) shows that the causal attention bias dominates the RoPE recency bias, enabling front‑positioned tokens to exert greater influence. Based on these findings, placing sink tokens at the front ensures they receive stronger, earlier attention, resulting in consistently better performance than appending them at the end.
>
> **Training-from-scratch**
>
> | Methods | GQA | SQA-image | TextVQA | MMMU | MME | MathVista |
> | :--- | :--- | :--- | :--- | :--- | :--- | :--- |
> | **Tinyllava 0.5B (Qwen)** | 58.0 | 57.8 | 47.3 | 30.3 | 1381.1 | 24.3 |
> | » Dual MLP with sink-to-the-end | 56.9 | 60.6 | 47.6 | 30.4 | 1433.9 | 24.6 |
> | » Dual MLP with sink-to-the-front | **58.3** | **60.6** | **48.4** | **30.8** | **1439.2** | **24.8** |
> | **Tinyllava 3B (Phi)** | 51.1 | 68.5 | 41.2 | 33.5 | 1455.2 | 25.9 |
> | » Dual MLP with sink-to-the-end | 49.8 | 65.9 | 37.6 | 34.3 | 1318.0 | 26.8 |
> | » Dual MLP with sink-to-the-front | **60.0** | **70.8** | **47.7** | **34.8** | **1679.2** | **28.4** |
>
> **Training-free**
> | Methods | GQA | SQA-image | TextVQA | MMMU | MME | MathVista |
> | :--- | :--- | :--- | :--- | :--- | :--- | :--- |
> | **InternVL2.5-4B** | 62.1 | 96.2 | **73.5** | 50.2 | 2332.2 | 62.9 |
> | ≫ sink-to-the-end | 62.2 | **96.2** | 73.3 | 50.3 | 2331.5 | 60.2 |
> | ≫ sink-to-the-front | **62.4** | 96.1 | 73.2 | **50.4** | **2351.3** | **63.3** |
> | **Phi 3.5 V** | 63.5 | 91.3 | **65.2** | 43.2 | 1887.9 | 43.1 |
> | ≫ sink-to-the-end | 63.5 | 91.0 | 64.2 | 43.8 | 1875.9 | 43.3 |
> | ≫ sink-to-the-front | **63.5** | **91.3** | 64.1 | **44.0** | **1891.3** | **43.5** |
>
>
> > Q8: What’s the "sink-to-the-front" performance for other VLMs, such as LLaVA models (e.g., TinyLLaVA-3B) ?
>
> **A**: In the paper, we applied the sink-to-the-front approach to four different LVLM variants. As shown in Q3 above, applying sink-to-the-front to Gemma-3-12B-it leads to improvements generally across benchmarks and especially on MME. Here, we provide additional training-free results for TinyLLaVA-3.1B-SigLIP-Phi2-3B, where we observe improvements on MathVista and comparable or slightly better performance on other benchmarks. Altogether, across six LVLM variants, we consistently observe positive gains, further supporting the robustness of our method.
>
> | Model                                   | **LLava_eval** |        |         |         | **MME** |        |        |         |         | **MathVista** |       |       |           |
> |-----------------------------------------|----------------|--------|---------|----------|-------------|----------------------|--------|--------|------------|-----------|---------|---------------|-------|
> |                                         | GQA   | SQA-image | TextVQA | MMMU-val | ALL         | ComRea | NumCal | TextTrans | CodeRea | ALL     | ALG     | GEO   | LOG   |
> | Tinyllava 3B (Phi)             | 51.09 | 68.47     | 41.15   | 33.50    | 1455.22     | 99.29  | 55.00  | 50.00     | 57.50   | 25.90   | 23.49   | 19.67 | 18.92 |
> | Sink-to-the front (Ours) | 51.08 | **68.47** | 38.97   | **33.90**| 1454.06     | **99.29** | **55.00** | **50.00** | **57.50** | **26.30** | **23.84** | **20.50** | **18.92** |

---

> ### Author Response · Authors · 2025-11-25
> **Reference**
>
> [1] Lee, B.K., Kim, C.W., Park, B. and Ro, Y.M., 2024. Meteor: Mamba-based traversal of rationale for large language and vision models. Advances in Neural Information Processing Systems, 37, pp.40278-40315.
>
> [2]  Wang, F., Wang, J., Ren, S., Wei, G., Mei, J., Shao, W., Zhou, Y., Yuille, A. and Xie, C., 2025. Mamba-Reg: Vision Mamba Also Needs Registers. In Proceedings of the Computer Vision and Pattern Recognition Conference (pp. 14944-14953).
>
> [3] Huang, W., Pan, J., Tang, J., Ding, Y., Xing, Y., Wang, Y., Wang, Z. and Hu, J., 2024. Ml-mamba: Efficient multi-modal large language model utilizing mamba-2. arXiv preprint arXiv:2407.19832.
>
> [4] Qiao, Y., Yu, Z., Guo, L., Chen, S., Zhao, Z., Sun, M., Wu, Q. and Liu, J., 2024. Vl-mamba: Exploring state space models for multimodal learning. arXiv preprint arXiv:2403.13600.
>
> [5] Chen, H., Song, Y., Chen, S., GE, C. and Huang, K., Unleashing the Power of Selective State Space Models in Vision-Language Models.
>
> [6] Huang, W., Pan, J., Wang, Z., Liu, Y., Wang, Y., Shen, S. and Hu, J., Enhancing Multimodal Large Language Models with Efficient Feature Alignment and Processing Using State Space Models. Available at SSRN 5294878.
>
> [7] Pantazopoulos, G., Nikandrou, M., Suglia, A., Lemon, O. and Eshghi, A., 2024. Shaking up vlms: Comparing transformers and structured state space models for vision & language modeling. arXiv preprint arXiv:2409.05395.
>
> [8] Zou, J., Liao, B., Zhang, Q., Liu, W. and Wang, X., 2025. Omnimamba: Efficient and unified multimodal understanding and generation via state space models. arXiv preprint arXiv:2503.08686.
>
> [9] Li, Z., Zhu, G., Yi, D. and Wang, J., 2024, October. Multimodal Mamba: A Versatile Multimodal Model for Seamless Integration into Diverse Downstream Tasks. In Proceedings of the 2024 13th International Conference on Computing and Pattern Recognition (pp. 303-313).
>
> [10] Xing, Y., Lan, X., Wang, R., Jiang, D., Huang, W., Zheng, Q. and Wang, Y., 2024. EMMA: Empowering Multi-modal Mamba with Structural and Hierarchical Alignment. arXiv preprint arXiv:2410.05938.
>
> [11] Zhao, H., Zhang, M., Zhao, W., Ding, P., Huang, S. and Wang, D., 2025, April. Cobra: Extending mamba to multi-modal large language model for efficient inference. In Proceedings of the AAAI Conference on Artificial Intelligence (Vol. 39, No. 10, pp. 10421-10429).
>
> [12] Eom, S., Shim, J., Koo, G., Na, H., Hasegawa-Johnson, M.A., Kim, S. and Yoo, C.D., 2024, November. Query-based Cross-Modal Projector Bolstering Mamba Multimodal LLM. In Findings of the Association for Computational Linguistics: EMNLP 2024 (pp. 14158-14167).
>
> [13] Jaegle, A., Gimeno, F., Brock, A., Vinyals, O., Zisserman, A. and Carreira, J., 2021, July. Perceiver: General perception with iterative attention. In International conference on machine learning (pp. 4651-4664). PMLR.
>
> [14]  Su, J., Lu, Y., Pan, S., Murtadha, A., Wen, B. and Roformer, Y.L., 2023. Enhanced transformer with rotary position embedding., 2021. DOI: https://doi. org/10.1016/j. neucom.
>
> [15] Wu, X., Wang, Y., Jegelka, S. and Jadbabaie, A., 2025. On the emergence of position bias in transformers. arXiv preprint arXiv:2502.01951.
>
> [16] Wang, Z., Zhang, H., Li, X., Huang, K.H., Han, C., Ji, S., Kakade, S.M., Peng, H. and Ji, H., Eliminating position bias of language models: A mechanistic approach, 2025. URL https://arxiv. org/abs/2407.01100.

---

### Official Review · Reviewer_yLnB · 2025-10-27

**Soundness:** 4
**Presentation:** 3
**Contribution:** 3
**Rating:** 8
**Confidence:** 3

**Summary:**

The paper studies how large vision-language models (LVLMs) process visual information through looking into the *visual sink tokens.*

These are vision transformer (ViT) tokens with unusually large activation norms that have large attention score in the language model. The authors show that such tokens carry meaningful global scene information. They analyze how these sink tokens propagate into the language model, identify distinct hidden dimensions they activate, and demonstrate that they encode high-level semantics like object categories.

The paper also proposes two simple methods to improve LVLM performance with its findings:
(1) a simple inference-time trick that reorders sink tokens to appear earlier, and
(2) a training-based design called DIYSink that uses dual-MLP projectors for sink and non-sink tokens. Both improve multimodal reasoning performance on benchmarks like LLaVA eval, MME and MathVista.

**Strengths:**

1. It offers the first detailed analysis of how ViT sink tokens propagate into LVLMs and interact with the LLM.

2. The paper presents clear qualitative and quantitative evidence showing that sink tokens encode global, scene-level semantics, while non-sink tokens focus on local, fine-grained details.

3. Building on these findings, the authors propose two improvement methods: a training-free Sink-to-the-Front trick and a Dual-MLP design. Both of them are intuitive, simple-yet-effective, compatible with existing models, and lead to consistent gains.

4. This paper provide a more in-depth understanding to model interpretability and how the vision branch work in current LVLMs.

**Weaknesses:**

1. The threshold used to distinguish sink and non-sink tokens is manually selected and not adaptive. This choice is heuristic and its robustness is not fully verified. The method lacks an adaptive or learnable way to determine the threshold cutoff, which limits its generalization across models and datasets.
2. The experiments use CLIP and SigLIP-based LVLMs, all built on ViT encoders. It remains unclear if similar sink behavior and method effectiveness hold in other architectures like Mamba-based vision models that also show high-norm token patterns.

**Questions:**

The connection between this work and previous LVLM token sink study like [1] is not discussed. The paper could better clarify how its findings extend and differ from this work.

[1] Kaduri, O., Bagon, S., & Dekel, T. What’s in the image? A deep-dive into the vision of vision language models. CVPR, 2025.

---

> ### Author Response · Authors · 2025-11-25
> **Official Comment by Authors (1/3)**
>
> > Q1: Can the author verify the robustness of the selected threshold cutoff for sink and non-sink tokens to assure generalization across models and datasets? Can the author provide an adaptive way to determine the threshold cutoff for sink and non-sink tokens?
>
> **A: Summary/TL;DR: The sink-token threshold τ is a stable property of the visual encoder (e.g., CLIP, SigLIP), not dependent on the dataset or LLM. Empirically, token-norm distributions show a clear gap (e.g., 3-digit -> 2-digit), and sensitivity tests across thresholds (80–120) produce <0.4% variation, confirming robustness. To avoid manual tuning, as suggested, we also attempt an adaptive method that detects the last significant norm drop and evaluates that the appropriate threshold is selected across four LVLMs.**
>
> We appreciate the reviewer's concern regarding the manual selection of the sink token threshold, and our response addresses both its robustness and generalization. First, we would like to recall that the threshold τ is intrinsic to the visual backbone rather than the specific dataset or LLM. As long as the visual encoder remains consistent (e.g., CLIP or SigLIP), the characteristic distribution of token norms, and thus the threshold, remains stable across different models and datasets. For our train-from-scratch models using CLIP-ViT and SigLIP backbones, we empirically observed a distinct gap where sink token norms drop from 3-digit values (e.g., >150) to 2-digit values (e.g., ~47). Based on this, we set τ=100. To verify the robustness of this choice, we conducted sensitivity analyses by training DIYSink with CLIP+Qwen2-0.5B and SigLIP+Qwen2-0.5B using a range of thresholds (80, 90, 100, 110, 120). The results showed minimal performance variation, with MME score confidence intervals remaining below 0.2% for the CLIP-based model and 0.4% for the SigLIP-based model, confirming that the method is robust to threshold selection within a reasonable range.
>
> To further address the limitation of manual tuning and enhance generalization across various architectures, we have studied an adaptive, data-driven method for sink token selection. Based on the characteristics of sink tokens, we know that if the visual tokens are sorted by their norm, a significant relative drop in norm will be observed between the last sink token and the first non-sink token. Thus, we can identify sink tokens by detecting the last significant relative drop (e.g., greater than 10%) among the top-ranked tokens, allowing the model to adaptively determine the number of sink tokens for each model and dataset. We evaluated this adaptive method across four Large Vision-Language Model (LVLM) architectures, including training-from-scratch models like TinyLlava-0.5B (Qwen) and TinyLlava-3B (Phi), and training-free models like InternVL2.5-4B and Phi3.5V. The results on the MME benchmark show that this adaptive method achieves performance comparable to the fixed-threshold baseline, thereby proving its effectiveness by automating token selection without relying on a hard threshold. We will include a detailed discussion of this adaptive method in the appendix.
>
> | Methods | ALL | Perception | Cognition | ComRea | NumCal | TextTrans | CodeRea |
> | ----- | ----- | ----- | ----- | ----- | ----- | ----- | ----- |
> | | | | | | | | |
> | **Training-from-scratch** | | | | | | | |
> | **Tinyllava 0.5B (Qwen2)** | 1381.10 | 1173.96 | 207.14 | 82.14 | 37.50 | 50.00 | 37.50 |
> | DIYSink (CoT) hard | 1456.78 | 1179.28 | 277.50 | 85.00 | 60.00 | 80.00 | 52.50 |
> | DIYSink (CoT) adaptive | 1455.77 | 1183.27 | 272.50 | 85.00 | 55.00 | 80.00 | 52.50 |
> | | | | | | | | |
> | **Tinyllava 3B (Phi)** | 1455.22 | 1193.43 | 261.79 | 99.29 | 55.00 | 50.00 | 57.50 |
> | DIYSink (CoT) hard | 1523.18 | 1242.47 | 280.71 | 110.71 | 52.50 | 67.50 | 50.00 |
> | DIYSink (CoT) adaptive | 1526.01 | 1236.01 | 290.00 | 115.00 | 55.00 | 67.50 | 52.50 |
> | | | | | | | | |
> | **Training-free** | | | | | | | |
> | **InternVL2.5-4B** | 2332.16 | 1686.81 | 645.35 | 142.86 | 124.99 | 200.00 | 177.50 |
> | + Sink-to-the-front (hard) | 2351.33 | 1706.33 | 645.00 | 145.00 | 125.00 | 200.00 | 175.00 |
> | + Sink-to-the-front (adaptive) | 2357.18 | 1712.18 | 645.00 | 147.23 | 125.00 | 200.00 | 172.50 |
> | | | | | | | | |
> | **Phi3.5V** | 1887.94 | 1475.76 | 412.14 | 137.14 | 90.00 | 87.50 | 97.50 |
> | + Sink-to-the-front (hard) | 1891.27 | 1457.35 | 433.92 | 136.42 | 97.50 | 87.50 | 112.50 |
> | + Sink-to-the-front (adaptive) | 1889.28 | 1453.17 | 436.11 | 136.42 | 95.00 | 87.50 | 112.50 |

---

> ### Author Response · Authors · 2025-11-25
> **Official Comment by Authors (2/3)**
>
> > Q2: Does similar sink behavior and method effectiveness hold for LVLMs with other architectures such as Mamba-based vision models?
>
> **A: Summary/TL;DR: We have not found any LVLMs that use a Mamba-based visual encoder, although Mamba does appear in LVLMs as a connector or as the language model (e.g., Meteor [1]). DIYSink improves performance on Meteor [1], an LVLM that uses a Mamba connector and Mamba-based LLM.**
>
> We appreciate the reviewer’s concern and are happy to provide additional information. To the best of our knowledge, based on our survey of prior work [3–12], no recent LVLM employs a Mamba-based model as the visual encoder; ViT-based encoders such as CLIP and SigLIP remain the dominant architecture. Nevertheless, there have been some explorations of using Mamba as the connector or as the language model. To test how our method performs on such models, we applied our training-free method to Meteor [1] and observed performance improvements, as shown in the table below.
> Moreover, although we haven’t found existing LVLMs employ a Mamba-based ViT, prior work [2] has shown that sink tokens also emerge in Mamba based visual encoder. This suggests that DIYSink has the potential to naturally extend to future LVLMs that adopt Mamba-style visual encoders, further broadening its applicability.
>
>
> | Method | GQA | SQA-image | TextVQA | MMMU-val | MME |  |  |  |  | MathVista |  |  |  |
> |---|---|---|---|---|---|---|---|---|---|---|---|---|---|
> | | | | | | ALL | ComRea | NumCal | TextTrans | CodeRea | ALL | ARI | GEO | LOG |
> | **Meteor Baseline** | 62.54 | 94.29 | 67.01 | 39.00 | 2188.75 | 140.71 | 105.00 | 160.00 | 102.50 | 52.80 | 52.97 | 42.68 | 13.51 |
> | **Sink-to-the-front (Ours)** | **62.57** | **94.65** | **67.07** | **39.10** | **2198.47** | **141.42** | **105.00** | **160.00** | **110.00** | **53.00** | **53.54** | **42.68** | **13.51** |
>
> ---
> **Reference**
>
> [1] Lee, B.K., Kim, C.W., Park, B. and Ro, Y.M., 2024. Meteor: Mamba-based traversal of rationale for large language and vision models. Advances in Neural Information Processing Systems, 37, pp.40278-40315.
>
> [2]  Wang, F., Wang, J., Ren, S., Wei, G., Mei, J., Shao, W., Zhou, Y., Yuille, A. and Xie, C., 2025. Mamba-Reg: Vision Mamba Also Needs Registers. In Proceedings of the Computer Vision and Pattern Recognition Conference (pp. 14944-14953).
>
> [3] Huang, W., Pan, J., Tang, J., Ding, Y., Xing, Y., Wang, Y., Wang, Z. and Hu, J., 2024. Ml-mamba: Efficient multi-modal large language model utilizing mamba-2. arXiv preprint arXiv:2407.19832.
>
> [4] Qiao, Y., Yu, Z., Guo, L., Chen, S., Zhao, Z., Sun, M., Wu, Q. and Liu, J., 2024. Vl-mamba: Exploring state space models for multimodal learning. arXiv preprint arXiv:2403.13600.
>
> [5] Chen, H., Song, Y., Chen, S., GE, C. and Huang, K., Unleashing the Power of Selective State Space Models in Vision-Language Models.
>
> [6] Huang, W., Pan, J., Wang, Z., Liu, Y., Wang, Y., Shen, S. and Hu, J., Enhancing Multimodal Large Language Models with Efficient Feature Alignment and Processing Using State Space Models. Available at SSRN 5294878.
>
> [7] Pantazopoulos, G., Nikandrou, M., Suglia, A., Lemon, O. and Eshghi, A., 2024. Shaking up vlms: Comparing transformers and structured state space models for vision & language modeling. arXiv preprint arXiv:2409.05395.
>
> [8] Zou, J., Liao, B., Zhang, Q., Liu, W. and Wang, X., 2025. Omnimamba: Efficient and unified multimodal understanding and generation via state space models. arXiv preprint arXiv:2503.08686.
>
> [9] Li, Z., Zhu, G., Yi, D. and Wang, J., 2024, October. Multimodal Mamba: A Versatile Multimodal Model for Seamless Integration into Diverse Downstream Tasks. In Proceedings of the 2024 13th International Conference on Computing and Pattern Recognition (pp. 303-313).
>
> [10] Xing, Y., Lan, X., Wang, R., Jiang, D., Huang, W., Zheng, Q. and Wang, Y., 2024. EMMA: Empowering Multi-modal Mamba with Structural and Hierarchical Alignment. arXiv preprint arXiv:2410.05938.
>
> [11] Zhao, H., Zhang, M., Zhao, W., Ding, P., Huang, S. and Wang, D., 2025, April. Cobra: Extending mamba to multi-modal large language model for efficient inference. In Proceedings of the AAAI Conference on Artificial Intelligence (Vol. 39, No. 10, pp. 10421-10429).
>
> [12] Eom, S., Shim, J., Koo, G., Na, H., Hasegawa-Johnson, M.A., Kim, S. and Yoo, C.D., 2024, November. Query-based Cross-Modal Projector Bolstering Mamba Multimodal LLM. In Findings of the Association for Computational Linguistics: EMNLP 2024 (pp. 14158-14167).

---

> ### Author Response · Authors · 2025-11-25
> **Official Comment by Authors (3/3)**
>
> > Q3: Can the authors discuss the connection and difference between your paper and the prior work [1]?
>
> **A: Summary/TL;DR: We are happy to clarify the difference between our paper and Deep-Dive into the Vision of VLMs (Kaduri et al.) [1].  Although both papers analyze information pathways in LVLMs, as summarized in the table above, they study fundamentally different token types, and thus reveal different functional roles of the tokens,  and with different focuses.**
>
> As summarized in Table 1 below, Deep-Dive [1] analyzes LLM-emerged sink tokens, which are created inside the LLM, while we focus on ViT sink tokens, which are high-norm visual tokens pre-summarized by the expert image encoder. Deep-Dive [1]  finds that LLM-emerged sinks capture global perceptual cues useful for object localization and image captioning, whereas our analysis shows that ViT sinks encode high-level abstract semantics that support global reasoning tasks. Building on their findings, Deep-Dive [1] proposes efficient image-reprompting to maintain performance on MME perception tasks using fewer visual tokens. In contrast, DIYSink goes beyond efficient inference and focuses on leveraging ViT tokens dynamically to achieve broader and more consistent improvements across benchmarks. We also like to point out that with the baseline model, inputting only ViT sink tokens to the LLM can yield a more efficient and better-performing model on multiple reasoning tasks, as shown in Table 2 below.
>
>
> **Table 1: Comparison Between Deep-Dive [1] and DIYSink**
>
> | Aspect                     | Deep-Dive [1]                                                         | DIYSink (Ours)                                                                 |
> |---------------------------|-----------------------------------------------------------------------|---------------------------------------------------------------------------------|
> | **Token Type Studied**    | LLM-emerged sink tokens                                               | ViT sink tokens               |
> | **Where Tokens Originate**| Inside the LLM during text decoding                                   | In the ViT before reaching the LLM                                             |
> | **What the Tokens Encode**| Global perceptual cues useful for localization and captioning          | High-level abstract semantics supporting global reasoning                     |
> | **Main Contribution / Method** | Image-reprompting for efficient inference with fewer visual tokens | Dynamic use of ViT tokens for broader and more consistent improvements        |
> | **Focus**                 | Improving efficiency on perception tasks (e.g., MME)                  | Improving reasoning and overall multimodal performance                          |
> | **Additional Finding**    | —                                                                     | Using only ViT sink tokens improves efficiency and reasoning performance (Table 2) |
>
> **Table 2: Using only ViT sink tokens improves efficiency and reasoning performance**
>
> | Methods               | **MME**        |            |          | **MathVista** |        |        |        |
> |-----------------------|----------------|------------|----------|---------------|--------|--------|--------|
> |                       | NumCal         | TextTrans  | CodeRea  | ALG           | GEO    | SCI    | GPS    |
> | TinyLLaVA-0.5B (Qwen2) | 37.50          | 50.00      | 37.50    | 22.42         | 20.92  | 36.89  | 22.12 |
> | **Baseline Inference w/ sink only** | **50.00**     | **65.00**  | **50.00** | **25.27**     | **23.85** | **38.52** | **25.96** |
> | LLaVA 7B | 55.00          | 55.00      | 42.50    | 25.27         | 20.92  | 39.34  | 21.15 |
> | **Baseline Inference w/ sink only** | **65.00**    | **95.00**  | **65.00** | **32.03**     | **30.13** | **43.44** | **34.13** |
>
>
> ---
> **Reference**
>
> [1] Kaduri, O., Bagon, S., & Dekel, T. What’s in the image? A deep-dive into the vision of vision language models. CVPR, 2025.

---

### Official Review · Reviewer_UaWB · 2025-10-29

**Soundness:** 3
**Presentation:** 3
**Contribution:** 3
**Rating:** 6
**Confidence:** 3

**Summary:**

This paper shows that ViT “attention sinks” in LVLMs are not noise but compact carriers of global semantics that propagate into the LLM with distinct activation patterns. Leveraging this, the authors propose a training-free token reordering (sink-to-the-front) and a trainable DIYSink with dual MLP projection and dynamic selection; both yield consistent gains, especially on global reasoning tasks, while sinks can hurt highly local tasks.

**Strengths:**

1. The paper proposes both a training-free and a training-based approach (Sink-to-the-Front, DIYSink) to utilize sink tokens. These methods are simple yet effective, model-agnostic, and easy to integrate into existing LVLM pipelines without retraining the full model.

2. The discovery that only 1–3% of ViT tokens (the sinks) dominate semantic propagation implies a strong potential for visual token compression and efficient inference. Even though not the main focus, this insight could inspire future research on adaptive token selection and efficient LVLM decoding.

**Weaknesses:**

1. The efficiency implications of identifying only 1–3% ViT sink tokens are not explored; analyzing their potential for token pruning or lightweight inference would enrich the contribution.
2. The task taxonomy (global, local, mixed) depends entirely on GPT-4o annotation without human verification, raising concerns about the consistency of the main conclusion.
3. The experimental coverage is relatively limited, lacking evaluation on hallucination-oriented (e.g., POPE) and high-resolution (e.g., V*, HRBench) benchmarks, which would better demonstrate generality.

**Questions:**

1. Given that only about 1–3% of visual tokens are identified as ViT sinks, have the authors considered using this finding for token pruning or efficient inference? For example, could a “sink-only inference” configuration preserve comparable performance while significantly reducing FLOPs and memory cost?

2. The task taxonomy (global, local, mixed) is annotated by GPT-4o. Was any human verification or inter-annotator check performed?

3. The current evaluation mainly includes structured QA benchmarks (MME, MathVista). Could the authors consider extending experiments to hallucination-oriented benchmarks such as POPE, and high-resolution benchmarks such as V* or HRBench, to further validate the generality of the proposed method?

---

> ### Author Response · Authors · 2025-11-25
> **Official Comment by Authors (1/2)**
>
> We sincerely appreciate the thoughtful feedback from Reviewers qH3u, UaWB, yLnB, ZDRY, and 4Rra, as well as their constructive suggestions on how to improve the paper. We are deeply encouraged that reviewers found our findings novel and interesting (ZDRY, qH3u, 4Rra) and valuable to the community (qH3u, 4Rra). Reviewers also noted that our paper is well-structured with clear motivation (qH3u), offers thorough analysis (yLnB, ZDRY), provides sufficient qualitative and quantitative evidence to support our findings (yLnB), and presents a sound approach and application (qH3u, 4Rra); our proposed methods incur low engineering overhead (4Rra) and are intuitive, simple yet effective, practical, and easy to integrate into existing LVLMs (ZDRY, yLnB, UaWB); our paper provide comprehensive experiments across diverse models and benchmarks (qH3u), showing consistent performance gains (yLnB), and recognized the strong potential for visual token compression and efficient inference suggested by our work (UaWB). We greatly appreciate the time reviewers spent evaluating our paper and offering insightful comments. We address each concern in detail in our responses, and we would be happy to clarify any additional questions during the discussion phase.
>
> **Rebuttal**
>
> > Q1: Can the author explore the efficiency implications of identifying only 1–3% ViT sink tokens?
>
> **A: Summarly/TL;DR: Sink tokens can support efficient, lightweight inference. Using only the tiny set of ViT sink tokens (≈0.5% of visual tokens) improves performance on multiple high-level reasoning tasks.**
>
> We thank the reviewer for this insightful suggestion. As shown in the table above, we examine the efficiency implications of using only ViT sink tokens and observe consistent improvements on high-level global reasoning tasks. Specifically, we run inference in the baseline models by feeding only the ViT sink tokens into the LLM (0.5% of the visual tokens), and we see performance gains in Numerical Calculation, Text Translation, and Code Reasoning of MME; on MathVista, we observe improvements in Algebraic Reasoning, Geometry Reasoning, Scientific Reasoning, and Geometry Problem Solving.
>
> These results reinforce our finding that ViT sink tokens encode highly abstract, global visual information. Because these tokens capture the distilled semantics most relevant for global reasoning, they enable efficient inference with improving performance on reasoning-centric benchmarks. We will include these findings in the revision to better highlight their implications for token pruning and lightweight inference.
>
> | Methods| **MME**||| **MathVista** ||        |        |
> |--|----|-|----------|---------------|--------|--------|--------|
> || NumCal| TextTrans  | CodeRea  | ALG           | GEO    | SCI    | GPS    |
> | **TinyLLaVA-0.5B-Qwen2** | 37.50          | 50.00      | 37.50    | 22.42         | 20.92  | 36.89  | 22.12 |
> | baseline Inference w/ sink only | **50.00**     | **65.00**  | **50.00** | **25.27**     | **23.85** | **38.52** | **25.96** |
> | **LLaVA 7B** | 55.00          | 55.00      | 42.50    | 25.27         | 20.92  | 39.34  | 21.15 |
> | baslline Inference w/ sink only | **65.00**    | **95.00**  | **65.00** | **32.03**     | **30.13** | **43.44** | **34.13** |
>
>
> > Q2: Can the author provide human verification on the task taxonomy with GPT-4o?
>
> **A: Summary/TL;DR: Our human validation study confirms that GPT-4o’s task classifications exhibit reliability comparable to that of human annotators. GPT-4o achieves a Kappa score of 0.526 and 0.4267 as compared to 0.6897 and 0.5263 for humans, respectively.**
>
> We thank the reviewer for raising the consistency concern, and providing the practical suggestion. To address the concern regarding the consistency of our GPT-4o-based task taxonomy, we conducted a human verification study involving 12 evaluators who assessed 84 VQA pairs using the same criteria provided to GPT. To first get a  reliability ceiling for this subjective task, we had two different people annotate the same set of 14 VQA pairs. This yielded Cohen's Kappa scores of 0.690 (Image Complexity) and 0.583 (Query Globalness), indicating that even human experts reach only "substantial" rather than "near perfect" agreement. Against this human baseline, GPT-4o demonstrated robust alignment. When comparing the continuous decimal scores using a 0.5 tolerance, machine judgement achieved Kappa scores of 0.526 (moderate agreement) and 0.427 (moderate agreement), respectively, and a combined correlation coefficient of 0.512 (compared to 0.667 for humans). These results indicate that GPT captures the underlying difficulty distribution with a reliability approaching that of human annotators.
>
> | | Image Complexity | Query Globalness | Correlation Coefficient |
> | :--- | :--- | :--- | :--- |
> | **Human vs Human** | 0.6897 | 0.5826 | 0.6667 |
> | **Human vs GPT** | 0.5263 | 0.4267 | 0.512 |

---

> ### Author Response · Authors · 2025-11-25
> **Official Comment by Authors (2/2)**
>
> >Q3. Additional results on other types of benchmarks, such as hallucination-oriented tasks (e.g., POPE), for demonstrating generality?
>
> We thank the reviewer for suggesting additional benchmarks to test the generalization of our approach. As suggested, we have extended our evaluation to include the POPE (hallucination-oriented) benchmark. Our results on POPE shows that our training-from-scratch models consistently improve over the baseline. We attribute this to our method's ability to dynamically reweight toward local non-sink tokens. Since POPE focuses on object existence (a more localized task), this reweighting allows the model to better attend to specific object details. The results are shown below. As for our training-free method, we observe performance on par with the baseline. Note that as POPE is more of a local task, that does not heavily rely on global context, the benefits of using sink tokens are less pronounced in this scenario.
>
> | Model Name | POPE |
> | :--- | :--- |
> | **Tinyllava 0.5B (Qwen2)** | 86.80 |
> | Our (CoT) | **87.15** |
> | Our (ReW) | 86.10 |
> | **Tinyllava 3B (Phi)** | 84.00 |
> | Our (CoT) | **86.13** |
> | Our (ReW) | 85.90 |
> | **Tinyllava-3B (Qwen2.5)** | 85.93 |
> | Our (CoT) | **86.33** |
> | Our (ReW) | 86.10 |
> | **LLaVA 7B** | 87.16 |
> | Our (CoT) | 86.80 |
> | Our (ReW) | **87.30** |

---

### Official Review · Reviewer_qH3u · 2025-11-01

**Soundness:** 4
**Presentation:** 3
**Contribution:** 4
**Rating:** 8
**Confidence:** 4

**Summary:**

This paper mainly investigates the ViT sink phenomenon in the MLLM domain. The main contributions are: (1) providing many insightful observations about the ViT sink and its underlying mechanisms, and (2) designing applications that leverage these findings to enable MLLMs to route between non-sink and sink states under different scenarios, which is both useful and insightful.

**Strengths:**

* Valuable observations and insights for the multimodal large language model community.

* The overall logical flow of the paper is well-structured — from motivation, to investigation methods, to applications.

* Comprehensive experiments are conducted across different models and benchmarks.

* Some findings are quite interesting. For example: “Moreover, these high-value sink dimensions emerge only after multimodal training. In LLaVA-7B, the LLM’s original sink dimensions are {2533, 1415}, while the propagated ViT sink tokens activate dimensions {982, 2494, 3263}.”

**Weaknesses:**

* Some parts are confusing and not well presented or clearly clarified.

  In Section 3.2, some results show that specific head sinks may correspond to either the foreground or the background, while other results indicate that all sink tokens map to the main object of the image. The authors also conclude that ViT sinks capture coarse-grained, high-level contextual features. However, identifying a small main object in a large image seems more like a fine-grained task. I am a bit confused — could the authors provide a clearer and more unified explanation of these observations?

* Some representations could be further improved.

  In the current Figure 3, the leftmost image is not easily readable and should be better presented. A simple improvement would be to split it into two sub-images instead of keeping it as one.

**Questions:**

* For an image token, the positional embedding indicates its location on the image. However, repositioning these tokens breaks this property. Although the results are promising, I would like to hear the authors’ explanation of this phenomenon.

* The proposed two-step chain-of-thought process could also be redesigned to use a soft rather than hard selection mechanism. For example, one could model the probability that a given question requires holistic reasoning.

* Additionally, are the two introduced MLP parameters summed to equal the original one? If not, the comparison may not be entirely fair.

---

> ### Author Response · Authors · 2025-11-25
> **Official Comment by Authors (1/2)**
>
> We sincerely appreciate the thoughtful feedback from Reviewers qH3u, UaWB, yLnB, ZDRY, and 4Rra, as well as their constructive suggestions on how to improve the paper. We are deeply encouraged that reviewers found our findings novel and interesting (ZDRY, qH3u, 4Rra) and valuable to the community (qH3u, 4Rra). Reviewers also noted that our paper is well-structured with clear motivation (qH3u), offers thorough analysis (yLnB, ZDRY), provides sufficient qualitative and quantitative evidence to support our findings (yLnB), and presents a sound approach and application (qH3u, 4Rra); our proposed methods incur low engineering overhead (4Rra) and are intuitive, simple yet effective, practical, and easy to integrate into existing LVLMs (ZDRY, yLnB, UaWB); our paper provide comprehensive experiments across diverse models and benchmarks (qH3u), showing consistent performance gains (yLnB), and recognized the strong potential for visual token compression and efficient inference suggested by our work (UaWB). We greatly appreciate the time reviewers spent evaluating our paper and offering insightful comments. We address each concern in detail in our responses, and we would be happy to clarify any additional questions during the discussion phase.
>
> **Rebuttal**
> > Q1: Can the authors provide more information for Section 3.2. 1) Why Figure 4 B shows that specific head sinks may correspond to either the foreground or the background while Figure 4 C shows that all sink tokens map to the main foreground object of the image. 2) Why sinks capture coarse-grained, high-level contextual features while Figure 4 B shows examples involving small objects.
>
> **A: Summary/TL;DR: 1) Section 3.2 Figure 4 C is showing the word distribution of H12, which is a foreground head. 2) “Coarse-grained” here refers to semantic role rather than object size. Sink tokens highlight the scene’s main semantic entity, whether large or small, and they have a lower capacity for capturing fine-grained local details, such as specific object parts.**
>
> We appreciate the reviewer’s thoughtful question and the opportunity to clarify this point.
>
> Section 3.2 Figure 4 C is showing the word distribution weighted by relevancy map from H12, which is a foreground head and thus illustrates to main objects. We would make sure to clarify that in the revised paper.  We did not visualize the background head because the word distribution of sink would map to a wide variety of tokens describing the different backgrounds of the 300 images we used to plot the distribution.
>
> Our main finding for section 3.2 is that ViT sinks capture coarse-grained, high-level contextual information aligned with the specific focus of each attention head.  Here, our notion of coarse-grained, high-level information is not tied to the physical size of an object in the image, but rather to the functional role of the object within the visual scene. For the foreground head, whether the main dominant object occupies a large or small portion of the image, identifying it reflects scene-level semantics (the type of information typically used for tasks like image-level classification), rather than fine-grained, part-level information.
>
> In the examples we provided in Figure 4 B, the sink-relevance maps consistently highlight the most salient object in the scene. This object may be visually small or large, but in either case, is representative of the dominant semantic entity, which is precisely the type of high-level, coarse contextual information that ViT sinks capture. In contrast, intricate local details, such as textures, object parts, or fine spatial relationships, are less likely what sink tokens tend to encode.
>
> > Q2: The leftmost image of Figure 3 is not easily readable. Split it into two sub-images.
>
> **A:** We appreciate the reviewer’s comment. As suggested, we had enlarged the images and also splited into two sub figures in the appendix Figure A10.

---

> ### Author Response · Authors · 2025-11-25
> **Official Comment by Authors (2/2)**
>
> > Q3: Will repositioning visual tokens breaks the spatial layout of images?
>
> **A：Summary/TL;DR: Our reordering doesn’t break spatial structure because we adjust the positional embedding accordingly. We included the discussion into Sec. B3 and Fig. A2.**
>
> *Training-free*: as noted in Line 323 of the main paper, we preserve the spatial structure by moving both the sink tokens and their original positional embeddings together. This ensures that the spatial layout encoded by the position embeddings remains intact, even though the tokens are reordered in the sequence.
>
> *Train-from-scratch*: the model is retrained with sink tokens copied to the front, while their original positions are filled with constant vectors. This preserves the overall spatial grid and allows the model to learn to interpret the modified sequence without losing spatial consistency. Illustraction is provided in Fig. A2.
>
> Importantly, sink tokens typically correspond to only 3–5 background tokens out of a much larger set of visual tokens (often 576 or more). Because they constitute such a small fraction of the sequence, repositioning them has minimal impact on the effective spatial layout of the image. This also helps explain why models remain stable despite the reordering.
>
> > Q4: Can we redesign the proposed two-step chain-of-thought process to a soft rather than hard selection mechanism?
>
> **A: Summary/TL;DR: We kindly note that Reweighting MLP in Section 4.2 is a soft selection mechanism. Following the reviewer’s idea, we additionally converted the CoT procedure into a soft selector, which performs on-par with the original CoT.**
>
> We thank the reviewer for proposing an additional selection mechanism. First, we kindly recall that in Section 4.2, we proposed both a Chain-of-Thought (CoT) approach and a learnable reweighting (ReW) mechanism. The reweighting mechanism is a soft selection mechanism, assigning soft values for weighting the sink and non-sink tokens.
>
> Following the reviewer's suggestion, we redesigned the CoT into a soft selection mechanism for evaluation. Specifically, for a given task, we first ran the initial 10 VQA samples using the standard CoT to obtain the empirical distribution of sink and non-sink token usage. For instance, in a code reasoning task, if the first ten samples used sink-only inference (6 samples), non-sink-only inference (2 samples), and both (2 samples), the resulting soft weighting values are calculated as $(6+2, 2+2)/(6+2+2) = (0.67, 0.33)$. We then reran the inference for the code reasoning task using $(0.67, 0.33)$ as the soft reweighting values. The results show that the performance of the soft CoT and the original CoT are on-par. We observed that while the overall performance of the soft CoT is slightly higher than the original CoT, the performance on the cognition task can drop slightly.
>
> |Methods|ALL|Perception|Cognition|ComRea|TextTrans|CodeRea|
> |:-|:-|:-|:-|:-|:-|:-|
> |**Tinyllava 0.5B (Qwen2)**|1381.10|1173.96|207.14|82.14|50.00|37.50|
> |DIYSink (ReW) | 1451.87 | 1222.23 |229.64|87.14|50.00|37.50|
> |DIYSink (CoT)|**1456.78**|1179.28|**277.50**|85.00|**80.00**|**52.50**|
> |DIYSink (Soft-CoT) | 1456.32 | **1238.46**|217.86 |**87.86** | 50.00 | 32.50 |
> |||||||||
> |**Tinyllava 3B (Phi)** | 1455.22 | 1193.43 | 261.79 | 99.29 | 50.00 |**57.50**|
> |DIYSink (ReW)|**1682.41**|**1416.70**|265.71| 115.71 |57.50| 45.00 |
> |DIYSink (CoT)|1523.18|1242.47|**280.71** | 110.71  | **67.50**| 50.00 |
> |DIYSink (Soft-CoT)|1579.85|1310.21|269.64|**117.14**| 50.00 | 47.50 |
>
> > Q5:  Are the two introduced MLP parameters summed to equal the original one?
>
> **A: Summary/TLDR: The Dual-MLP module introduces only about 0.2% additional parameters. We demonstrate that this marginal increase is not the source of performance gains by testing a baseline with a doubled MLP size, which performed on par with the original.**
>
> We appreciate the reviewer’s concern regarding the parameter count. While the dual-MLP does double the parameters of the original MLP, the connector constitutes only a very small portion of the overall LVLM. For example, in TinyLLaVA-0.5B, our dual-MLP module adds only about 0.2% additional parameters (1.8M out of ~1B), making it unlikely that the observed improvements are driven solely by parameter increase.
>
> To further disentangle this effect, we also evaluated a baseline with doubled MLP depth. Its performance is comparable to, or slightly above, the original baseline, yet it consistently remains below DIYSink. This suggests that our gains stem from the design itself rather than from simply adding more parameters.
>
> |Model|LLaVA Eval|MME ALL|MathVista ALL|
> |-|-|-|-|
> | TinyLLaVA 0.5B (Qwen2.5)| 49.57| 1506.68   | 24.40|
> | TinyLLaVA 0.5B (Qwen2.5) w/ doublemlp| 50.18| 1500.95   | 25.10|
> | Our (CoT)| 50.02| 1522.45| 26.00|
> | Our (ReW)| **50.34**| **1526.69**| **26.90**|

---

### Public Comment · ~Amil_Dravid1 · 2025-11-14
**Great Paper! Might want to check out our work**

This is a great paper! The claims "ViT attention sinks – a problem that has been rarely studied but is indeed very important for LVLMs" and "training-free and training-based approaches to better leverage how this information is interpreted by the LLM" make it seem pretty related to our work: https://arxiv.org/abs/2506.08010, which studies the mechanism behind sinks in ViTs, offers a training-free intervention, and also has some small experiments in VLMs. While we were able to get some better text-to-vision attribution in VLMs, we weren't able to really improve VLM benchmarks, so your work is exciting! Would be interested to hear how your results connect to some of our findings in that paper.

---

### Author Response · Authors · 2025-11-26
**To all Reviewers**

We sincerely appreciate the thoughtful feedback from Reviewers qH3u, UaWB, yLnB, ZDRY, and 4Rra, as well as their constructive suggestions on how to improve the paper. We are deeply encouraged that reviewers found our findings novel and interesting (ZDRY, qH3u, 4Rra) and valuable to the community (qH3u, 4Rra). Reviewers also noted that our paper is well-structured with clear motivation (qH3u), offers thorough analysis (yLnB, ZDRY), provides sufficient qualitative and quantitative evidence to support our findings (yLnB), and presents a sound approach and application (qH3u, 4Rra); our proposed methods incur low engineering overhead (4Rra) and are intuitive, simple yet effective, practical, and easy to integrate into existing LVLMs (ZDRY, yLnB, UaWB); our paper provide comprehensive experiments across diverse models and benchmarks (qH3u), showing consistent performance gains (yLnB), and recognized the strong potential for visual token compression and efficient inference suggested by our work (UaWB). We greatly appreciate the time reviewers spent evaluating our paper and offering insightful comments. We address each concern in detail in our responses, and we would be happy to clarify any additional questions during the discussion phase.

In this rebuttal, we provide:
- **Expanded Model Architectures**: Additional experiments on LVLMs using larger backbones (Gemma-12B), state-space model components (Mamba), and cross-attention connectors (Q-Former, Perceiver Resampler).
- **Hallucination Analysis**: New performance results on the POPE benchmark.
- **Sink Token Mechanics**: Clarification on the positional embedding of sink tokens, including qualitative results justifying the choice to pre-pend (move to front) rather than append (move to end) the tokens.
- **Efficiency Analysis**: Discussion on the inference efficiency gains when leveraging sink tokens for high-reasoning tasks.
- **Evaluation Validity**: A human study validating the alignment of our GPT-based evaluation results.
- **Robustness Checks**: Additional discussion on the robustness of threshold selection and the use of adaptive methods.
- **Differentiation**: A detailed comparison clarifying the differences between our method and Deep-Dive.
- **Statistical Significance**: Statistical analysis confirming the reliability of our model improvements.

---

### Meta-Review · Area_Chair_KtGB · 2026-01-05

**Summary:**

This paper studies for the first time the beneficial effects of visual sinks on the performance of VLMs and, based on it proposes, strategies to employ them effectively to improve the performance of VLMs. The work is insightful and novel. Five reviewers, provided mostly positive reviews, with most of their major concerns being effectively addressed by the authors' rebuttal, including those of the one negative reviewer that provided an initial rating of 2. The latter would have likely increased their score after the rebuttal.

All things considered, the AC feels that the paper makes a novel and significant contribution to the research literature and recommends acceptance. The authors should include all the changes that they have promised in the rebuttal into their final camera ready version.

Congratulations!

**Reviewer Concerns:**

Concerns addressed:
* Evaluation on larger models > 7B
* Evaluation on Mamba-based ViTs connectors
* Providing statistical significance of the improvements in accuracy
* Many additional clarifications of the method and its positioning and theoretical motivation
* Efficacy of the method for pruning and efficiency
* Evaluation of the position of visual sink tokens

Concerns not addressed:
* Evaluation on HRBench
* Evaluation on image encoders not using ViT as no SOTA VLMs are based on non-ViT architectures

**Reviewer Scores:**

1. Reviewer qH3u (Rating: 8: accept, good paper (poster))

This reviewer's primary concerns were around seeking further clarification about the methodology and some additional experiments. The authors' rebuttal addresses most of their concerns. They would have likely maintained their positive rating.

2. Reviewer UaWB (Rating: 6: marginally above the acceptance threshold. But would not mind if paper is rejected)

This reviewer raised three main concerns: (a) testing the utility of ViT sink tokens for efficiency and pruning; (b) human verification of tasks categorized into global/local/both by ChatGPT-4o and (c) evaluations of the proposed approach on POPE and HRBench. The authors provided additional experiments to address each of these concerns. The would have likely maintained or increased score.

3. Reviewer yLnB (Rating: 8: accept, good paper (poster))

Their main concerns were (a) that the threshold used to distinguish sink and non-sink tokens is manually selected and not adaptive, (b) evaluation on Mamba-based image encoder models and (c) connections of the work to the prior work called "Deep-Dive". The authors provide numerical results to address the first two concerns and also provide a clear discussion of the distinctions of their work to Deep-Dive. The reviewer is likely to have maintained their original positive score.

4. Reviewer ZDRY (Rating: 2: reject, not good enough)

The reviewer's main concerns were (a) the lack of theoretical grounding, (b) only marginal improvements in accuracy; (c) limited evidence on larger models > 7B, (d) generalization beyond ViTs (CNN, Mamba, etc) and (e) Incremental contribution (shows only empirical observation). The authors provided detailed numerical experiments to sufficiently address all these concerns. The reviewer would have likely increased their score.

5. Reviewer 4Rra (Rating: 8: accept, good paper (poster))

This reviewer's primary concerns were (a) why moving the sink tokens to the front helps and (b) evaluation on larger models > 7B. The authors addressed both these concerns adequately. They would have likely maintained their positive rating.

---

### Decision · Program_Chairs · 2026-01-26

Accept (Poster)